# Persistence of alveolar fibroblast-derived ADAMTS4+ cells in a preclinical model of delayed pulmonary fibrosis resolution

Mahsa Zabihi[1,2,3,13], Ali Khadim [1,2,3,13], Arun Lingampally [1,2,3,4] ✉,
Ana Ivonne Vazquez-Armendariz[5], Stefan Hadzic[2,3,4],
Georgios-Dimitrios Panagiotidis[1,2,3,4], Daniel Kalina[2,3,4,6], Jan Halweg[1,2,3],
Tara Procida-Kowalski[2,3,7], Marek Bartkuhn [2,3,7], Xuran Chu[8,9],
Janine Koepke[2,3,4], Christos Samakovlis [2,3,4], Mario Boehm[6],
Norbert Weissmann[2,3,4], Andreas Günther [2,3,4], Werner Seeger [2,3,4],
Peter Braubach [10], Susanne Herold [1,2,3], Malgorzata Wygrecka [2,3,4,6],
Saverio Bellusci[11,12] ✉ & Elie El Agha [1,2,3,8] ✉

Idiopathic pulmonary fibrosis is the most common and aggressive form of interstitial lung disease. Despite extensive research on the pathomechanisms of fibrogenesis, little is known about the mechanisms of fibrosis resolution. Here, lineage tracing of alveolar fibroblasts was carried out during fibrosis development and delayed resolution in aged mice. Histological analyses, single-cell transcriptomics, and ex vivo models including alveolar organoids and precision-cut lung slice cultures were employed. The data reveal that lipofibroblasts contribute to myofibroblast formation during fibrogenesis, with the reverse differentiation trajectory occurring during fibrosis resolution. Importantly, delayed resolution is associated with the persistence of ADAM metallopeptidase with thrombospondin type 1 motif 4-positive (ADAMTS4+) cells. Investigation of human lung transplant tissues, single-cell and spatial transcriptomic datasets, and functional ex vivo interventions reveal strong clinical relevance. Our study underscores the significance of the lipofibroblast-to-myofibroblast reversible switch in fibrosis development and resolution and identifies ADAM metallopeptidase with thrombospondin type 1 motif 4 as a potential therapeutic target in human lung fibrosis.

Idiopathic pulmonary fibrosis (IPF) is widely perceived as an aberrant wound-healing response resulting from repetitive microinjuries to the alveolar epithelium and continuous scarring of the alveolar network. This pathological course features excessive deposition of extracellular matrix (ECM) proteins and culminates in progressive loss of gas exchange and ultimately respiratory failure. IPF is the most common and relentless form of interstitial lung disease (ILD), with a median survival of 3–5 years after diagnosis[1–3]. Aging, environmental

exposures, and genetic factors are associated with an increased risk of developing IPF[4–6]. This disease typically manifests in middle-aged adults, suggesting a strong link between aging and increased incidence and prevalence[7]. Hallmarks of aging including genomic instability, telomere shortening, cellular senescence, mitochondrial dysfunction, and elevated reactive oxygen species (ROS) have been implicated in IPF pathogenesis[8]. Notably, NADPH oxidase 4 (NOX4), an enzyme involved in ROS generation and regulation, is elevated in IPF lungs, and

saverio.bellusci@innere.med.uni-giessen.de; elie.el-agha@innere.med.uni-giessen.de

its inhibition has been shown to attenuate lung fibrosis[9]. Metformin, a well-known antidiabetic drug, has demonstrated antifibrotic effects via adenosine monophosphate-activated protein kinase (AMPK) signaling that inhibits NOX4[10] and induces apoptosis in myofibroblasts (MyoFBs)[11], the effector cells that promote fibrotic remodeling in the lung[12–14]. We previously showed that metformin inhibits collagen production in MyoFBs in an AMPK-dependent manner and induces lipogenic differentiation in an AMPK-independent manner[15].

Lipofibroblasts (LIFs) are alveolar fibroblasts located adjacent to alveolar type 2 (AT2) cells[16] These cells contain neutral lipid droplets and are characterized by the expression of adipose differentiation-related protein (*Adrp* a.k.a. perilipin 2 or *Plin2*)[17], transcription factor 21 (*Tcf21*)[18], and fibroblast growth factor 10 (*Fgf10*)[19,20]. It is important to mention that there is no single marker or transgenic line that can currently be used to exclusively label or target LIFs. We previously lineage-traced *Plin2*+ cells in young adult mice to show that upon bleomycin injury, these cells give rise to MyoFBs[21]. Interestingly during fibrosis resolution in young mice, a subset of MyoFBs can revert to the native LIF phenotype, restoring normal lung architecture[15,21]. However, this reversible switch is not seen in human IPF, leading to a progressive and irreversible state.

Further refinement of the LIF population enriched with stem cell antigen 1 (*Sca-1*)/*Fgf10* double-positive cells suggested a higher potential for this subpopulation to support AT2 growth in alveolar organoids[22]. However, aging impairs the niche activity of resident mesenchymal cells (rMCs) due to increased *Nox4* expression, which can be reversed by NOX4 inhibition[23]. Moreover, lipid scarcity in aging lungs has been implicated in reduced AT2 renewal capacity, and lipid replenishment has been shown to restore this activity in alveolar organoids[24].

Recent single-cell RNA sequencing (scRNA-seq) studies have highlighted the role of LIF-to-MyoFB differentiation during fibrosis formation[25–28]. These studies identified transitional cell states, such as secreted frizzled related protein 1-positive (*Sfrp1*+), RUNX family transcription factor 1-positive (*Runx1*+), lipocalin 2-positive (*Lcn2*+), and alveolar fibroblast 3 (AL3) subpopulations that link LIFs and collagen triple helix repeat containing 1 (*Cthrc1*+) MyoFBs[27,29,30]. Targeting these transitional states led to lung simplification and emphysema-like phenotypes thus emphasizing their importance in therapeutic strategies[30]. However, most of these studies focused on young mice, which is a limitation since bleomycin-induced pulmonary fibrosis in young mice spontaneously resolves, failing to replicate the persistent fibrosis observed in human IPF patients. Since IPF is an age-related disease, studying fibrosis in aged mice may provide more clinically relevant insights. Aged mice exhibit delayed fibrosis resolution and prolonged transitional states, making them a better model for understanding differentiation trajectories during both fibrosis formation and delayed resolution[31,32].

In this study, we used 52–to–56-week-old female *Fgf10*^*Cre-ERT2/+*; *tdTomato*^*flox* mice to model fibrosis with delayed resolution, mimicking human IPF pathology. We performed scRNA-seq on lineage-labeled *Fgf10*+ cells following bleomycin-induced lung injury to investigate cellular heterogeneity and fate trajectories. Additionally, we used alveolar organoid and precision-cut lung slice (PCLS) cultures to test potential therapeutic interventions. We finally validated our findings by investigating publicly available databases and employing fresh human lung tissues.

## Results

### Aged mice demonstrate incomplete fibrosis resolution following bleomycin injury

Female *Fgf10*^*Cre-ERT2/+*; *tdTomato*^*flox* mice at 52-to-56 weeks of age were administered three intraperitoneal (i.p.) injections of tamoxifen followed by a single i.t. instillation of bleomycin (Bleo) or saline (Sal) (Fig. 1a). This experimental approach enabled the labeling of pre-

existing *Fgf10*+ cells, allowing to track their fate during fibrosis formation and resolution. Lungs were harvested at day 14 (d14) (corresponding to the peak of fibrosis) and days 30 (d30) and 60 (d60) (corresponding to early and late fibrosis resolution, respectively) after the Bleo challenge (Fig. 1a). Hematoxylin and eosin (H&E) staining confirmed the development of pulmonary fibrosis by d14 where 40% ± 3.8% of the tissue was fibrotic (Supplementary Fig. 1a–c). Despite structural improvements from Bleo d30 till Bleo d60, persistent damage was still observed at these timepoints (24% ± 2.08% and 11.3% ± 1.4%, respectively) (Supplementary Fig. 1a–c). Masson's Trichrome and Picro Sirius Red staining showed increased collagen deposition at Bleo d14, and to a lesser extent at Bleo d30 and Bleo d60 (Supplementary Fig. 1d–g).

### Interconversion of *Fgf10*+ cells between the LIF and MyoFB states during fibrosis formation and resolution

We then assessed the status of LIF markers in Sal and Bleo lungs using RNAscope combined with immunofluorescence for surfactant protein C (SFTPC) (Fig. 1). In Sal lungs, 56.1% ± 6.3% of tdTom+ cells also expressed *Tcf21* and these cells were observed adjacent to SFTPC+ cells (AT2s) (Fig. 1b, k, l). Following bleomycin injury (Bleo d14), tdTom/*Tcf21* double positive cells and AT2s were dramatically lost in fibrotic regions (Fig. 1c, j–l). Similarly, tdTom/*Inmt*/*Robo2* triple-positive cells were decreased from 34.1% ± 2.6% in Sal to 7.09% ± 0.9% in Bleo d14 (Fig. 1f, g, m). These results suggest that tdTom+ cells lose their LIF profile at the peak of fibrosis. Conversely, during early (Bleo d30) and late (Bleo d60) fibrosis resolution, the number of AT2s and the expression of LIF markers in lineage-labeled cells were largely, yet incompletely, restored (Fig. 1d, e, h, i).

To characterize lineage-labeled *Fgf10*+ (tdTom+) cells and precisely determine their fate during fibrosis development and resolution, live, mesenchymal tdTom+ cells were FACS-sorted from Sal, Bleo d14, Bleo d30, and Bleo d60 lungs (Fig. 2a, b). FACS analysis demonstrated significant abundance of tdTom+ cells in Bleo d14 lungs and sustained presence of these cells during resolution (Bleo d30 and d60) (Fig. 2c). Next, sorted tdTom+ cells were loaded for scRNA-seq. Integrative UMAPs showed that tdTom+ cells in Sal predominantly consist of alveolar fibroblasts 1 (AF1 or LIF; around 55%) and AF2 (around 36%) along with peribronchial fibroblasts (PeriFBs; around 6.6%) and traces of myofibroblasts (MyoFBs; around 2.6%) (Fig. 2d, e). The top DEGs for each cluster are shown in Fig. 2f. In situ hybridization on Sal lung sections confirmed such heterogeneity at steady state and showed that tdTom+ cells contain (*Pi16*+) AF2, and (*Hhip*+) PeriFBs (Supplementary Fig. 2a–c) in addition to AF1s (Fig. 1). Importantly, the scRNA-seq analysis also showed that the MyoFB cluster peaked at Bleo d14 (7.3% of total cells) with partial regression during early (5.3% at Bleo d30) and late (3.9% at Bleo d60) resolution (Fig. 2d, e). Interestingly, the AF1 cluster expanded at the expense of AF2 during resolution (Fig. 2d, e).

To gain insights into differentiation trajectories, RNA velocity was carried out, and the results showed a clear transition from LIFs (AF1s) to MyoFBs at Bleo d14 and to a lesser extent at d30, while the reverse trajectory was observed at Bleo d60 (Fig. 2g). We then performed RNA velocity analysis with latent inferred time to further investigate differentiation trajectories, and the results showed a trajectory from a transitional state (*Sfrp1*+) to MyoFB/proliferative state and then LIFs (Fig. 2h). Notably, ADAM metallopeptidase with thrombospondin type 1 motif 4 (*Adamts4*) was among the top genes defining the terminal fate (Fig. 2h). Its expression was mainly observed in AF1s rather than MyoFBs, being induced at Bleo d14 and persisting at Bleo d60 (Fig. 2i–k and Supplementary Fig. 2d, e). ADAMTS4 was previously shown to mediate exuberant fibroblast activity leading to exaggerated immunopathology and lung damage after influenza virus infection[33] with one of its major substrates being versican (VCAN), which is an important immunomodulator that guides cell proliferation, migration, and differentiation during inflammation. ADAMTS4 is involved in maintaining

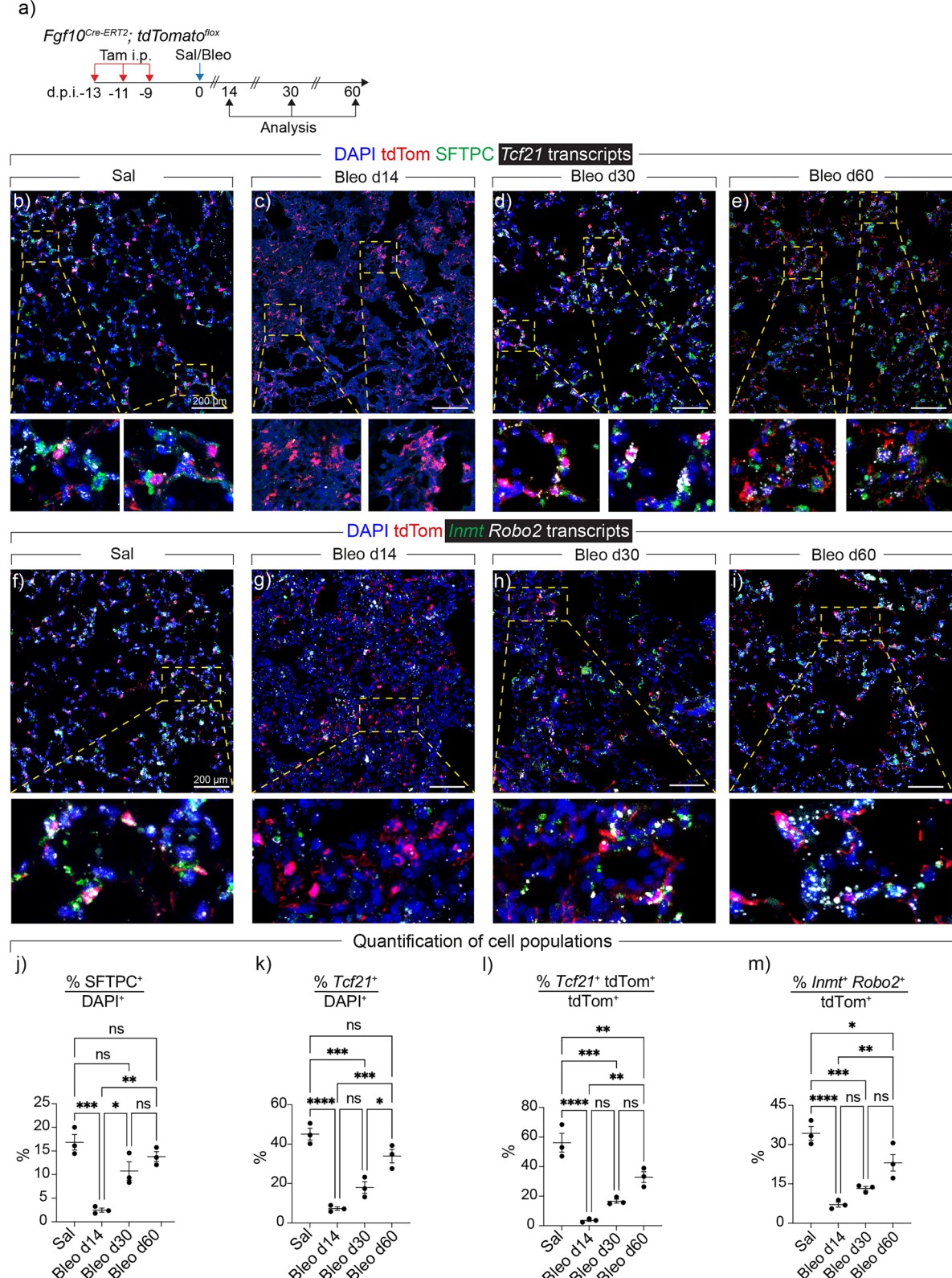

**Quantification of cell populations**

neutrophil infiltration and therefore sustaining an inflammatory milieu in the lung. We therefore hypothesized that the ADAMTS4/VCAN axis is involved in delayed resolution of pulmonary fibrosis.

In situ hybridization and immunofluorescence were carried out and the data revealed that tdTom+ cells were accumulated in fibrotic regions of Bleo d14 lungs (35.1% ± 2.7%) compared with normal regions of Sal lungs (19.77% ± 1.02%) (Fig. 3a, b, i). While the majority of tdTom+

cells were enriched with *Fgf10* expression in Sal (60.77% ± 3.78%) (Fig. 3a, k), only 6.3% ± 2.15% of tdTom+ cells expressed *Cthrc1* (Fig. 3e, o). At the peak of injury, tdTom+ cells lose *Fgf10* expression (Fig. 3b, k) and express *Cthrc1* (MyoFB marker) (37.1% ± 2%) (Fig. 3f, o). With the start of resolution, tdTom+ cells start reacquiring *Fgf10* expression and by Bleo d60, 38.8% ± 1.04% of them express *Fgf10* (Fig. 3c, d, k). In contrast, only 5.7% ± 1.24% *Cthrc1*/tdTom double-positive cells were

**Fig. 1 | Characterization of lipofibroblast marker expression in normal and fibrotic aged** *Fgf10<sup>Cre-ERT2</sup>; tdTomato<sup>flox</sup>* **lungs. a** Timeline and schematic of the experimental design. **b–e** Representative images of in situ hybridization for *Tcf21* (white) and immunofluorescence for pro-SFTPC (green) and tdTom (red). The dashed boxes are magnified in the lower panels. **f–i** Representative images of in situ hybridization for *Robo2* (white) and *Inmt* (green) and immunofluorescence for tdTom (red). The dashed boxes are magnified in the lower panels. **j–m** Quantification of in situ hybridization and immunofluorescence data. Scale bars: 200 μm. *n* = 3 per group. Each data point represents one biological replicate. Data are presented as mean ± SEM. Statistical analysis was performed using ordinary one-way ANOVA with Tukey's multiple comparisons test. * *P* < 0.05; ** *P* < 0.01; *** *P* < 0.001; **** *P* < 0.0001; ns Not significant. Bleo Bleomycin, DAPI 4′,6-diamidino-2-phenylindole, d.p.i. days post instillation, *Fgf10* Fibroblast growth factor 10, *Inmt* Indolethylamine N-methyltransferase, i.p. Intraperitoneal injection, *Robo2* Roundabout guidance receptor 2, Sal Saline, SFTPC Surfactant protein C, Tam Tamoxifen, *Tcf21* Transcription factor 21, tdTom tdTomato.

detected in the damaged regions of Bleo d60 (Fig. 3h, o). Further comparison of AF1s with MyoFBs revealed *Gsn, Inmt, C3, Igfbp6, Macf1* and *Clec3b* as top DEGs in AF1s and *Nrep, Postn, C1qtnf6* and *Ppic* in MyoFBs (Supplementary Fig. 2f). A heatmap showing bulk analysis of top DEGs in Sal, Bleo d14, Bleo d30, and Bleo d60 is shown in Supplementary Fig. 2g. The data showed a general switch to MyoFBs during fibrogenesis and to LIFs during resolution. The AF1 signature was highest in Bleo d60, which is in line with strong lipofibroblast differentiation and reinforcement during resolution (Supplementary Fig. 2g and Fig. Supplementary Fig. 3a). Bleo d14 was dominated by MyoFB and Bleo d30 showed gradual downregulation of MyoFB markers and induction of PeriFB and AF2 markers (Supplementary Figs. 2g and 3b). We also investigated the expression pattern of *Adamts4* by RNAscope and found that it is indeed upregulated in tdTom+ cells at d14 and persists at d60 (Fig. 3e–h, l, m, q). Its expression showed minimal overlap with *Cthrc1* (Fig. 3e–h, p), supporting the finding that *Adamts4* is largely expressed in AFs rather than MyoFBs during delayed resolution (Fig. 2i–k).

We also subclustered the MyoFB cluster to gain insights into the heterogeneity of these pathological cells and the transcriptomic changes occurring during fibrogenesis and recovery (Supplementary Fig. 4a, b). The analysis revealed seven subclusters including *Cthrc1<sup>high</sup>*, *Cthrc1<sup>low</sup>, Gli1<sup>high</sup>, Prolif<sup>high</sup>, LIF<sup>high</sup>, AF2<sup>high</sup>*, and a very minor SMC subcluster (Supplementary Fig. 4b, c). This was confirmed by applying a MyoFB (Supplementary Fig. 4d) and LIF scores (Supplementary Fig. 4e) reflecting the expression of the top 485 DEGs (Source Data file) for the respective populations. Analyzing individual timepoints showed the presence of almost only *LIF<sup>high</sup>* and *Cthrc1<sup>low</sup>* MyoFBs in Sal, while *Cthrc1<sup>high</sup>* and *Gli1<sup>high</sup>* MyoFBs were predominant in Bleo d14 before regressing through resolution (Supplementary Fig. 4f). The *AF2<sup>high</sup>* subcluster was prominent in Bleo d30 (Supplementary Fig. 4f). In Bleo d60, the *LIF<sup>high</sup>* cluster represented the majority of the remaining cells (Supplementary Fig. 4f), confirming our previous finding that MyoFB-to-LIF transition marks fibrosis resolution[15,21,34]. A heatmap of the top regulated genes in these subclusters is shown in Supplementary Fig. 4g. A similar analysis was conducted on the AF1 cluster and the analysis revealed ten subclusters including two *LIF<sup>high</sup>, LIF<sup>int</sup>*, two *LIF<sup>low</sup>*, *AF2<sup>high</sup>, AF2<sup>low</sup>, LIF<sup>Adamts4</sup>*, Apoptotic fibroblasts (*ApoptoticF*), and Proliferative fibroblasts (*ProlifF*) (Supplementary Fig. 4h–m). The two *LIF<sup>high</sup>* subclusters correspond to steady-state LIFs (red) and injury-induced LIFs (green). Injury-induced LIFs are enriched for regulation of apoptosis in response to DNA damage in the saline condition and programs such as ECM organization and remodeling and mesenchyme development during active fibrosis and resolution. Overall, the emerging picture from analyzing the AF1 cluster is that the *LIF* signature is dominant in Sal, lost during fibrogenesis, and re-emerges during late resolution (Bleo d60). Similarly to MyoFB subclustering, an *AF2* signature emerges during early resolution (Bleo d30) (Supplementary Fig. 4h, i, l). A heatmap of the top regulated genes in these subclusters is shown in Supplementary Fig. 4m.

### Inhibition of ADAMTS4 attenuates fibrogenesis ex vivo

To show a proof of concept that ADAMTS4 represents a potential therapeutic target and that its inhibition promotes fibrosis resolution, we carried out alveolar organoid assays by co-culturing WT alveolar epithelial cells (AECs) with *Sca1* + CD31/CD45/EpCAM triple-negative resident mesenchymal cells (rMCs) according to our previously published protocol[35]. Following sixteen days of culture, organoids were treated with recombinant transforming growth factor beta 1 (rTGFβ1) or vehicle (Veh) for three days followed by treatment with Veh or recombinant tissue inhibitor of metalloproteinase 3 (rTIMP-3) for three days (Fig. 4a). TIMP-3 is a known inhibitor of ADAMTS4 activity[36,37] although it also targets other metalloproteinases. Immunofluorescence and qPCR analysis showed that rTIMP3 treatment preserved organoid morphology (Fig. 4b–g), attenuated *Col1a1, Acta2,* and *Cthrc1* expression (Fig. 4h–j), and restored the expression of *Sftpc* (AT2 marker) and *Ager* (AT1 marker) (Fig. 4 m, n). Of note, *Adamts4* expression was induced by rTGFβ1, which was largely attenuated by rTIMP-3 treatment (Fig. 4k). The expression of endogenous *Timp3* was largely unchanged by the treatments (Fig. 4l).

As another layer of validation, we also generated PCLS from WT mice and subjected them to vehicle or rTGFβ1 treatment for five days. Since fibrogenesis is reversible in the PCLS model if rTGFβ1 is removed from the culture medium, PCLS were further treated with rTGFβ1 in the presence of vehicle or rTIMP-3 (Fig. 5a). Similarly to the organoid experiments, rTIMP3 treatment attenuated rTGFβ1-mediated fibrogenesis (Fig. 5b–d). Moreover, *Adamts4* per se was also downregulated upon rTIMP-3 treatment (Fig. 5e) while *Timp3* did not show significant changes (Fig. 5f). To add a functional dimension to these observations, protein lysates were subjected to an ADAMTS4 activity assay, and the results showed a significant increase in ADAMTS4 activity upon rTGFβ1 treatment, which was rescued by rTIMP-3 (Fig. 5g). To investigate the expression of ADAMTS4 in various mesenchymal subsets, thin sections were generated from the treated PCLS groups and RNAscope (for top markers of mesenchymal clusters) and immunofluorescence (for ADAMTS4) were carried out. The analysis confirmed the upregulation of ADAMTS4 at the protein level by rTGFβ1 treatment, and subsequent downregulation by rTIMP-3 (Fig. 5h–q). Moreover, the results showed that rTGFβ1 downregulates *Tcf21* (AF1 marker) (Fig. 5r) while inducing *Pi16* (AF2) (Fig. 5s), *Cthrc1* (MyoFB) (Fig. 5t), and to some extent *Hhip* (PeriFB) (Fig. 5u), which was largely reversed by rTIMP-3 treatment. Recombinant TGFβ1 induced the expression of ADAMTS4 mostly in AF1s (Fig. 5r) and AF2s (Fig. 5s), which was reversed by rTIMP-3, especially in AF1s. In another set of experiments, PCLS from aged Bleo d14 *Fgf10<sup>Cre-ERT2</sup>; tdTomato<sup>flox</sup>* mice were cultured ex vivo for three days while being treated with recombinant VCAN (rVCAN) or Veh, and analysis by qPCR, in situ hybridization, and immunofluorescence also showed attenuation of fibrosis and ADAMTS4 expression (Supplementary Fig. 5).

To validate our findings in the human context, we obtained lung explant tissues from IPF and non-IPF donors and carried out qPCR. The results showed significant upregulation of the MyoFB-associated genes *COL1A1, ACTA2, CTHRC1,* and *SPP1* (Fig. 6a) and downregulation of *SFTPC* and *HOPX* (as expected) (Fig. 6b), but also significant downregulation of AF1-associated genes *PDGFRA, CEBPB, LIMCH1, TCF21, INMT, PLIN2, PPARG,* and *NPNT* (Fig. 6c). We also observed downregulation of the endothelial marker *PECAM1* (Fig. 6d) and an upward trend for *ADAMTS4* (Fig. 6e). We also carried out RNAscope for *CTHRC1* and *ADAMTS4* on formalin-fixed, paraffin-embedded donor and IPF lung tissues and the data showed an increase in the abundance of *CTHRC1* + , *ADAMTS4*+, and double-positive cells in IPF lungs in parallel to a decrease in the abundance of cells

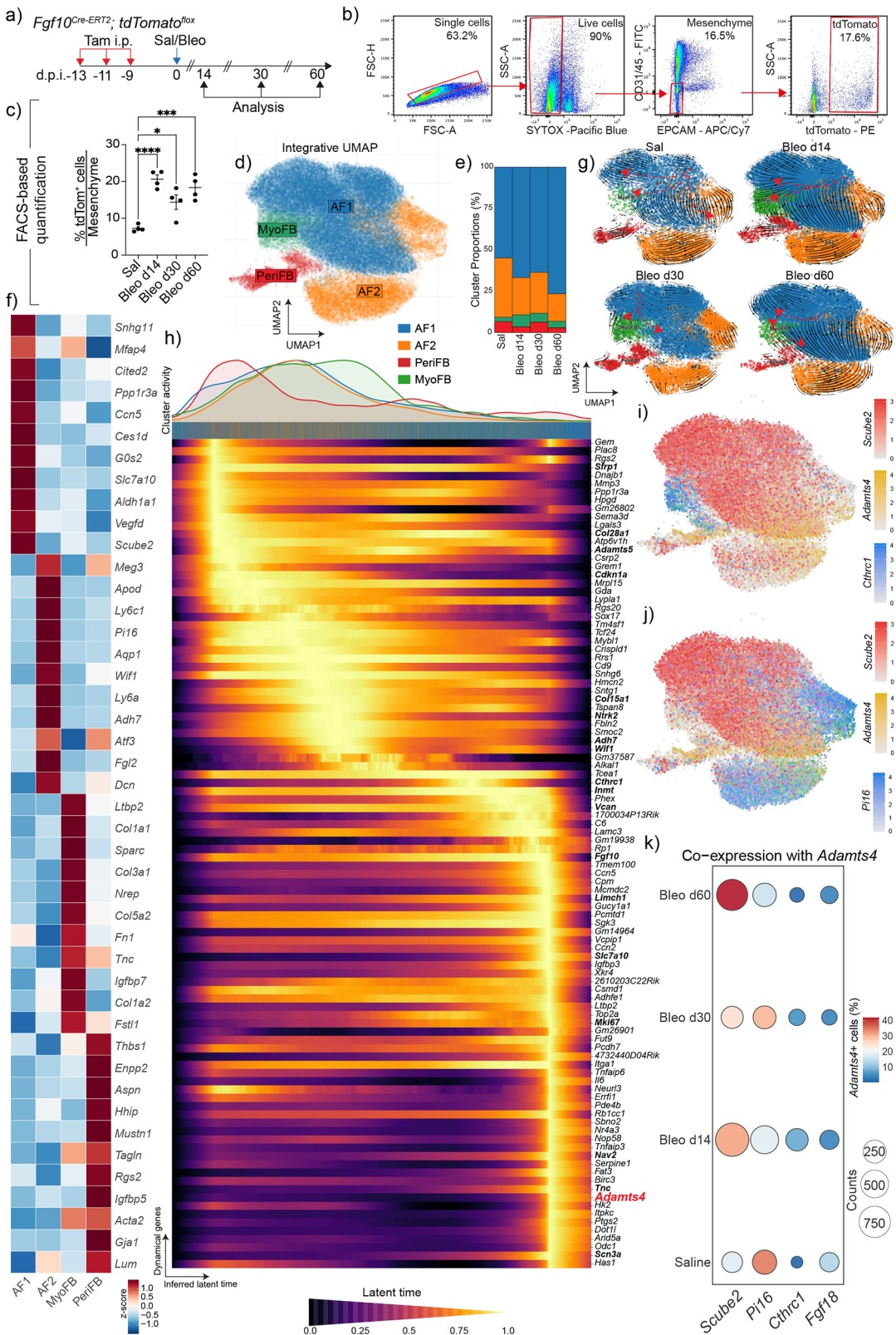

expressing the AF1 markers *INMT* and *ROBO2* (Fig. 6f–k). We also analyzed the human lung cell atlas (HLCA) by filtering for the stromal fraction in healthy, IPF, ILD, and non-specific interstitial pneumonia (NSIP) while preserving the original cluster annotation (Fig. 7a). Of note, cluster frequency analysis revealed significant loss of the AF1 cluster in IPF, ILD, and NSIP compared with the healthy state (Fig. 7b). We also carried out GSEA, and the results showed significant

downregulation of the *AF1* signature (defined in the Source Data file) in all fibrotic disease states (Fig. 7c). UMAP plots also showed downregulation of AF1 markers *FGF10, SCUBE2, INMT*, and *LIMCH1* in parallel to upregulation of *CTHRC1* and *ADAMTS4* (Fig. 7d). Importantly, interrogating the top 10 DEGs in an unbiased manner revealed that *ADAMTS4* is among the top genes that are upregulated in NSIP, IPF, and ILD compared with the healthy state (Fig. 7e). We also generated a

**Fig. 2 | Single-cell transcriptomics reveals the heterogeneity of the *Fgf10+* lineage in normal and fibrotic aged lungs. a** Timeline and schematic of the experimental design. **b** Gating strategy for isolating tdTom+ cells from the live mesenchymal population for single cell RNA sequencing (scRNA-seq). **c** Quantification of sorted lineage-labeled cells in saline and experimental groups. **d** Integrative Uniform Manifold Approximation and Projection (UMAP) plot of lineage-labeled *Fgf10+* cells isolated from Sal, Bleo d14, Bleo d30, and Bleo d60 showing different clusters. **e** Frequency of clusters in each condition. **f** Heatmap showing the top DEGs in each cluster. **g** RNA velocity analysis for individual timepoints. **h** Integrative RNA cell velocity with latent inferred time and the corresponding dynamical genes. **i, j** UMAP plots showing the expression patterns of indicated genes. **k** Dot plot showing co-expression of *Adamts4* and indicated genes.

Source data are provided as a Source Data file. In **c** n = 4 per group; each data point represents one biological replicate. Data are presented as mean ± SEM. Statistical analysis was performed using ordinary one-way ANOVA with Dunnett's multiple comparisons test. *$P < 0.05$; ***$P < 0.001$; ****$P < 0.0001$. *Adamts4* ADAM metallo-peptidase with thrombospondin type 1 motif 4, AF1 Alveolar fibroblasts 1, AF2 Alveolar fibroblasts 2, Bleo Bleomycin, d.p.i. days post instillation, EpCAM Epithelial cell adhesion molecule, *Fgf10* Fibroblast growth factor 10, *Hhip* Hedgehog interacting protein, i.p. Intraperitoneal injection, MyoFB Myofibroblasts, PeriFB Peribronchial fibroblasts, *Pi16* Peptidase inhibitor 16, Sal Saline, *Scube2* Signal peptide, CUB domain and EGF like domain containing protein 2, Tam Tamoxifen; tdTom: tdTomato.

volcano plot comparing IPF versus healthy, and *ADAMTS4* was among the top upregulated genes in IPF (Fig. 7f).

We then leveraged a published spatial transcriptomic dataset (S-BSST1410)[38] to interrogate the abundance and spatial distribution of various cell populations in donor and IPF (Fig. 8). As expected, we found significant loss of alveolar epithelial cells and accumulation of MyoFBs in IPF as indicated by downregulation of marker genes such as *NAPSA*, *SFTPC*, *AGER*, and *HOPX* in parallel to upregulation of genes related to collagen production and ECM remodeling (Fig. 8a, b). We also observed an overrepresentation of airway epithelial cells in IPF, indicating bronchiolization (Fig. 8a). Notably, *TIMP3* was among the top downregulated genes in IPF compared with donor (Fig. 8b). Spatial visualization of *CTHRC1* expression showed significant upregulation in the IPF samples, where its expression pattern mimicked that of MyoFBs (Fig. 8c vs. Fig. 8a). Importantly, *ADAMTS4* was also significantly upregulated in IPF samples (Fig. 8d, e), and there was a positive correlation between spots expressing *ADAMTS4* and those expressing *CTHRC1* in IPF (Fig. 8f), further confirming that *ADAMTS4* is upregulated in IPF and that its expression correlates with fibrogenesis. Collectively, the human single-cell and spatial transcriptomic data indicate that our findings using mouse-derived samples potentially possess strong clinical relevance in humans with pulmonary fibrosis.

Precision-cut lung slices were also generated from fresh IPF lung explants and treated with rTIMP-3 or rVCAN for three days (Fig. 9a). Gene expression analysis revealed a significant downregulation of *COL1A1* expression upon treatment (Fig. 9b). The treatment also strongly downregulated *ADAMTS4* (Fig. 9b) similarly to what was observed with mouse-derived samples (Fig. 5e and Supplementary Fig. 5b). Further histological analysis using in situ hybridization showed attenuation of *CTHRC1* and *ADAMTS4* in response to rTIMP-3 or rVCAN treatment while restoring *ROBO2* (AF1 marker) expression (Fig. 9c–f). Since rTIMP-3 is not a selective inhibitor of ADAMTS4 activity, siRNA-mediated knockdown of *ADAMTS4* was also carried out. Human PCLS derived from non-IPF donors were treated with Veh or rTGFβ1 for five days followed by similar treatment in the presence of scrambled siRNA or siRNA targeting *ADAMTS4* for three days (Fig. 9g). Immunofluorescence showed upregulation of ADAMTS4 at the protein level in response to rTGFβ1 treatment, which was attenuated by treatment with si-*ADAMTS4* (Fig. 9h). As another layer of validation, qPCR and western blotting were performed and the results showed significant upregulation of COL1 and an upward trend for ADAMTS4 in response to rTGFβ1 treatment, which was attenuated by *ADAMTS4* gene silencing (Fig. 9i–k). Further analysis confirmed that ADAMTS4 activity was enhanced by rTGFβ1 treatment, which was attenuated in si-*ADAMTS4*-treated samples (Fig. 9l). Finally, RNAscope confirmed that *ADAMTS4* and *CTHRC1* were upregulated by rTGFβ1 treatment, which was attenuated by si-*ADAMTS4* treatment (Supplementary Fig. 6a–f, h). Knockdown of *ADAMTS4* also restored *ROBO2* expression (Supplementary Fig. 6b–e, g). Collectively, the data suggest that ADAMTS4 represents a potential therapeutic target in human lung fibrosis.

## Discussion

Ever since lipid-droplet-containing fibroblasts were reported in the alveolar interstitium of neonatal rat lungs[39], there has been controversy regarding the function or even existence of these elusive cells, which were later dubbed "LIFs". Apart from their involvement in assisting AT2s with pulmonary surfactant production, LIFs are widely regarded as niche cells for AT2s[40] as they produce important growth factors such as FGF10 that is important for AT2 self-renewal and maintenance[41,42]. Of note, while LIFs were sometimes claimed to only exist in rodent lungs[43], their presence was later confirmed in the human lung[44]. Despite that, the term "LIFs" is still often ignored in single-cell atlases of the healthy and fibrotic lung where the corresponding cells are generally assigned the AF label. Earlier work had demonstrated that fibroblast-derived radiolabeled lipids that are encapsulated by PLIN2 are taken up by AT2s[17]. Due to their strong expression of *Plin2*, the latter had been used as a marker of LIFs. We previously demonstrated that LIFs are a source of MyoFBs in the fibrotic lungs of young mice[21]. In that study, we lineage-traced *Plin2+* cells in the context of bleomycin-induced pulmonary fibrosis and subsequent resolution. However, the advent of single-cell transcriptomics allowed the identification of more reliable markers of LIFs thus enabling to carry out lineage tracing and the downstream analysis in a more precise manner. *FGF10* is a reliable marker of AF1s as evident from single-cell atlases and multiple other studies, and the *LIF* signature is enriched in the AF1 fraction of AFs.

Against this background, we utilized our *Fgf10^{Cre-ERT2}* knock-in line[45] to lineage-trace pre-existing *Fgf10+* cells in the context of pulmonary fibrosis. Since bleomycin-induced pulmonary fibrosis spontaneously resolves in young mice while human IPF is progressive/irreversible, we utilized old mice in the current study to create a state of delayed fibrosis resolution that we believe better mimics the clinical setting. Our data clearly demonstrate that the *LIF^{high}* fraction of *Fgf10+* cells contributes to MyoFB formation during fibrosis development. These data are not only in line with our previous work using young mice[21], but also with more recent literature implicating AFs as a major source of MyoFBs in the fibrotic lung[25–27,34]. During resolution, MyoFBs undergo apoptotic clearance or deactivation. The conversion of MyoFBs into AFs with a *LIF* signature is in line with our previously published data involving spontaneous or enhanced (via metformin treatment) resolution in young mice[15,21,34] as well as other recent studies[26,29,30,34]. We also previously demonstrated downregulation of *PLIN2*, *CEBPA*, and *PPARG* in lung homogenates from IPF lung tissues compared with donors indicating reduced LIF abundance/differentiation[21]. In the current work, we expanded the LIF gene panel to other marker genes shown to constitute a signature for LIFs according to single-cell atlases, and the data showed significant downregulation at the lung homogenate level as well as downregulation of the *LIF* signature in the stromal fraction of various fibrotic lung diseases contained in the HLCA. Altogether, our aged fibrosis mouse model and human-derived data strongly indicate that LIFs are indeed impaired in pulmonary fibrosis.

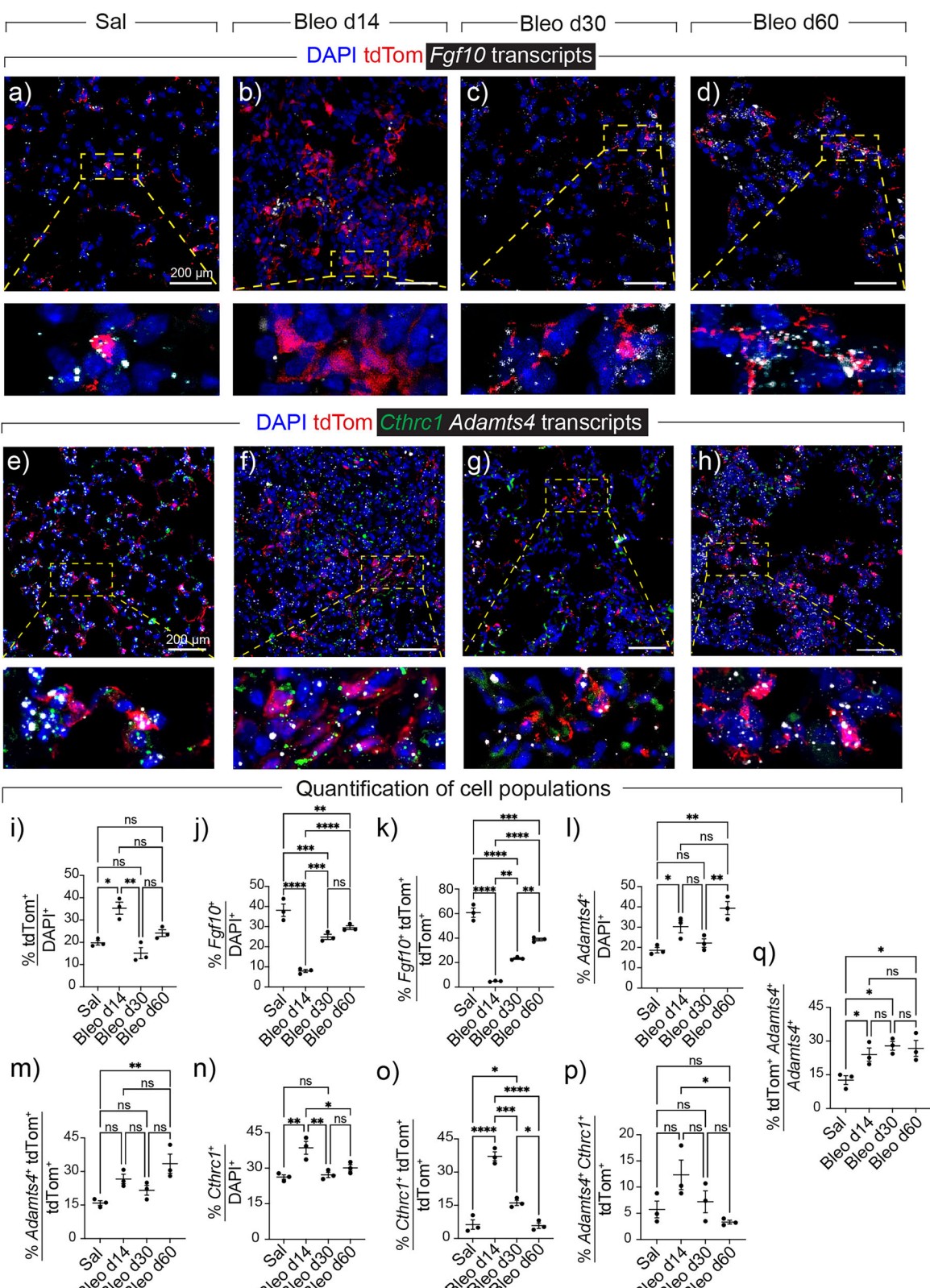

**Fig. 3 | Contribution of FGF10+ cells to myofibroblasts during fibrogenesis and persistent ADAMTS4+ cells during delayed resolution. a–d** Representative images of in situ hybridization for *Fgf10* (white) and immunofluorescence for tdTom (red) in saline and bleomycin-treated lungs. The dashed boxes are magnified in the lower panels. **e–h** Representative images of in situ hybridization for *Adamts4* (white) and *Cthrc1* (green) and immunofluorescence for tdTom (red) in saline and bleomycin-treated lungs. The dashed boxes are magnified in the lower panels. **i–q** Quantification of in situ hybridization and immunofluorescence data. Scale

bars: 200 μm. *n* = 3 per group. Each data point represents one biological replicate. Data are presented as mean ± SEM. Statistical analysis was performed using ordinary one-way ANOVA with Tukey's multiple comparisons test. *$P < 0.05$; **$P < 0.01$; ***$P < 0.001$; ****$P < 0.0001$; ns Not significant. *Adamts4* ADAM metallo-peptidase with thrombospondin type 1 motif 4, Bleo Bleomycin, *Cthrc1* Collagen triple helix repeat containing 1, DAPI 4′,6-diamidino-2-phenylindole, *Fgf10* Fibro-blast growth factor 10, Sal Saline, tdTom tdTomato.

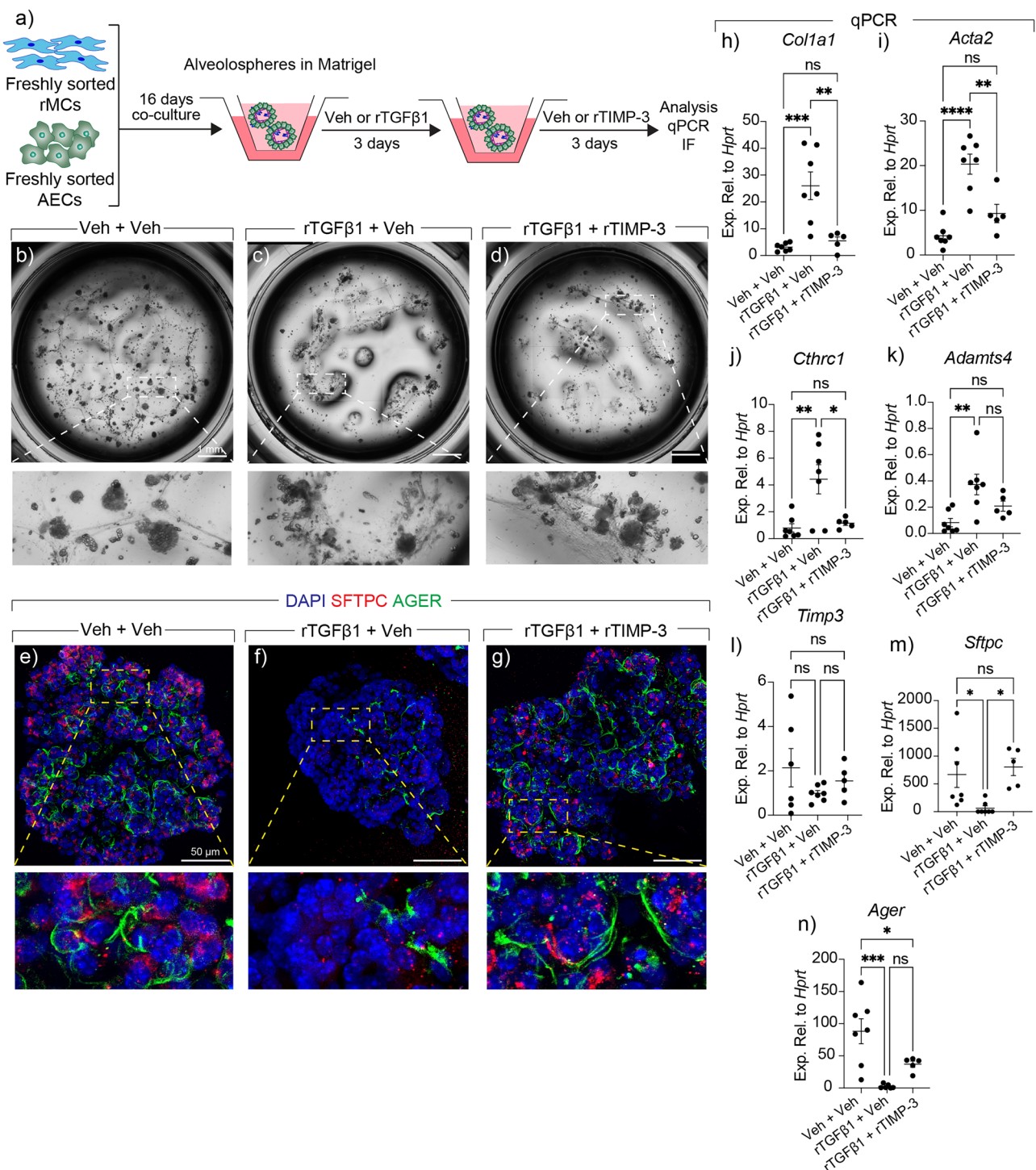

**Fig. 4 | Inhibiting ADAMTS4 mitigates fibrogenesis in murine alveolar organoids. a** Timeline and schematic of the experimental design. **b–d** Representative tile scans of wells from different experimental groups. The dashed boxes are magnified in the lower panels. **e–g** Representative confocal images of immunofluorescence for AGER (green) and SFTPC (red). **h–n** Quantitative PCR for the indicated genes. $n = 7$ for Veh + Veh (except for *Timp3* $n = 6$); $n = 7$ for rTGFβ1 + Veh (except for *Ager* $n = 6$); $n = 5$ for rTGFβ1 + rTIMP-3. Scale bars: **b–d** 1 mm; **e–g** 50 μm. Each data point represents one biological replicate. Data are presented as mean ±

SEM. Statistical analysis was performed using ordinary one-way ANOVA with Tukey's multiple comparisons test. *$P < 0.05$; **$P < 0.01$; ***$P < 0.001$; ****$P < 0.0001$; ns Not significant. *Acta2* Actin alpha 2, smooth muscle, *Adamts4* ADAM metallopeptidase with thrombospondin type 1 motif 4, AECs Alveolar epithelial cells, AGER Advanced glycosylation end-product specific receptor, DAPI 4′,6-diamidino-2-phenylindole, rMCs Resident mesenchymal cells, rTGFβ1 recombinant transforming growth factor beta 1, rTIMP-3 recombinant tissue inhibitor of metalloproteinases, SFTPC Surfactant protein C, Veh Vehicle.

Importantly, our preclinical model identified *Adamts4* + AFs that are associated with delayed fibrosis resolution. While bioinformatic analysis predicted that *Fgf10*+ cells do not necessarily pass through a MyoFB intermediate before giving rise to *Adamts4*+ cells, this

intriguing aspect remains an open question that requires further experimental testing. During homeostasis, the balance between ECM synthesis and degradation is crucial for maintaining proper lung physiology and architecture. However, such ECM homeostasis is disrupted

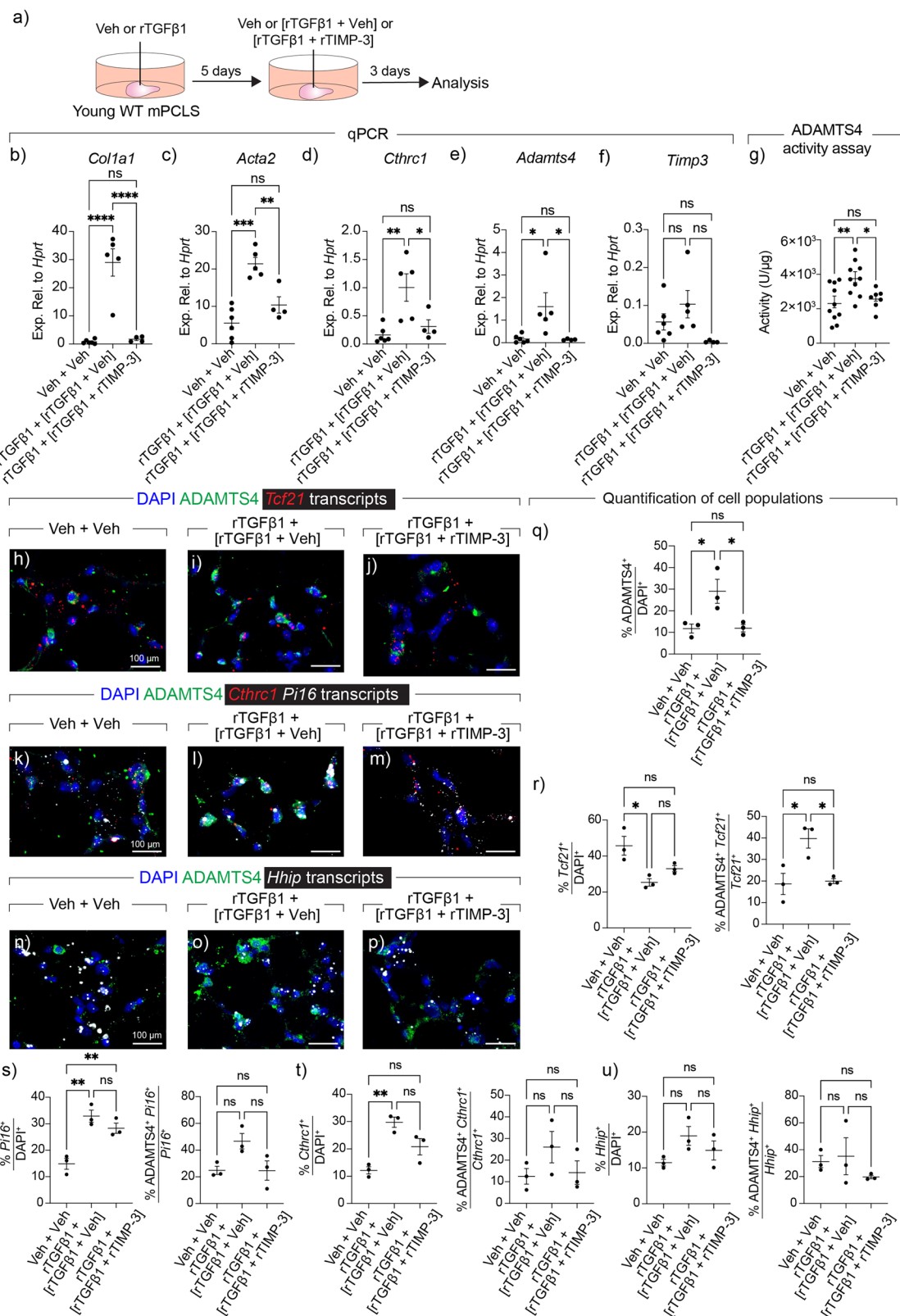

during lung fibrosis. Metalloproteinases play pivotal roles in regulating ECM turnover and fibrosis progression. MMPs exhibit both pro-fibrotic (for e.g., MMP-1, MMP-7, and MMP-9) and anti-fibrotic (e.g., MMP-2 and MMP-3) activities. Elevated levels of MMP-1, MMP-7, and MMP-9 have been observed in the blood serum and lung tissue of IPF patients, with MMP-7 considered a potential diagnostic biomarker[46–48]. Of note, ADAMTS4 has been implicated in lethal immunopathology following

influenza virus injury[33]. VCAN is a major substrate for ADAMTS4, and it is a component of the provisional matrix that facilitates recruitment of inflammatory cells such as leukocytes[33,49]. TIMP-3 inhibits ADAMTS4 but also other metalloproteinases. *Timp3* expression is significantly reduced during fibrosis formation in young mice but recovers during fibrosis resolution. *Timp3* knockout mice exhibit persistent inflammation and fibrosis, resembling the pathology observed in aged mice.

**Fig. 5 | Inhibiting ADAMTS4 mitigates fibrogenesis in murine precision-cut lung slice cultures. a** Timeline and schematic of the experimental design. **b–f** Quantitative PCR for the indicated genes. **g** ADAMTS4 activity assay. **h–p** Representative images of in situ hybridization for the indicated targets (red and white) and immunofluorescence for ADAMTS4 (green) in various experimental conditions. **q–u** Quantification of in situ hybridization and immunofluorescence data. **b–f** $n = 6$ for Veh + Veh; $n = 5$ for rTGFβ1 + Veh; $n = 4$ for rTGFβ1 + rTIMP-3. **g** $n = 10$ for Veh + Veh; $n = 10$ for rTGFβ1 + Veh; $n = 7$ for rTGFβ1 + rTIMP-3. **q–u** $n = 3$ per group. Scale bars: 100 μm. Each data point represents one biological replicate.

Data are presented as mean ± SEM. Statistical analysis was performed using ordinary one-way ANOVA with Tukey's multiple comparisons test. *$P < 0.05$; **$P < 0.01$; ***$P < 0.001$; ****$P < 0.0001$; ns Not significant. *Acta2* Actin alpha 2, smooth muscle, ADAMTS4 ADAM metallopeptidase with thrombospondin type 1 motif 4, *Cthrc1* Collagen triple helix repeat containing 1, DAPI 4′,6-diamidino-2-phenylindole, *Fgf10* Fibroblast growth factor 10, *Hhip* Hedgehog interacting protein, *Pi16* Peptidase inhibitor 16, rTGFβ1 recombinant transforming growth factor beta 1, rTIMP-3 recombinant tissue inhibitor of metalloproteinases, Veh Vehicle.

This effect is attributed to sustained MMP activity, increased neutrophil infiltration, and impaired anti-inflammatory responses. Datamining of spatial transcriptomic data revealed that *TIMP3* is among the top downregulated genes in IPF compared with donor. Interestingly, MMP-7 activity is elevated in *Timp3* knockout mice, while *Mmp7* knockout mice demonstrate impaired neutrophil migration and attenuated fibrosis after bleomycin injury[50,51]. Our data using rTIMP-3 or rVCAN in the context of alveolar organoids and PCLS revealed strong attenuation of fibrogenesis. As both candidates are not ADAMTS4-specific, we also employed a genetic approach and showed that siRNA-mediated knockdown of *ADAMTS4* yields similar antifibrotic effects. Importantly, datamining of the HLCA revealed that *ADAMTS4* is among the top upregulated genes in IPF, ILD, and NSIP compared with the healthy state. Spatial transcriptomic data also demonstrated the significant upregulation of *ADAMTS4* in IPF, and its correlation with fibrogenesis. These data strongly suggest that the relentless course of fibrogenesis seen in human pulmonary fibrosis may be due to continued inflammation and remodeling mediated by ADAMTS4 and VCAN degradation. Our ex vivo investigations revealed that rTIMP-3 or rVCAN treatment attenuates fibrogenesis at least in part by downregulating *Adamts4* at the RNA and protein levels. To our knowledge, this phenomenon has not been described in the literature so far. *Adamts4* downregulation might be due to a feedback mechanism involving cues originating from the ECM being modulated by rTIMP3 or rVCAN. As expected, activity assays showed reduced ADAMTS4 activity upon rTIMP-3 or siRNA-mediated *ADAMTS4* knockdown. It is also possible that excess rVCAN saturates the protease activity of ADAMTS4, thus attenuating its detrimental effect. Another possibility is that VCAN acts directly on mesenchymal cells similarly to what has been shown in the context of cardiac repair[52]. It might also be that applying excess amounts of intact VCAN dilutes the profibrotic effect of VCAN fragments generated by ADAMTS4-mediated degradation. Thorough analysis of ECM modulation by ADAMTS4, its regulators, and its substrates as well as the associated contribution to sustained lung damage is an important aspect for further research. It will also be important to screen for small molecule inhibitors to selectively inhibit ADAMTS4 in the context of fibrogenesis and resolution using murine and humanized models of pulmonary fibrosis.

To conclude, our lineage-tracing approach identified *Fgf10* + AFs with *LIF* signature as a cellular source of not only MyoFBs that arise during fibrogenesis but also *Adamts4*+ cells that persist during delayed fibrosis resolution. Datamining and ex vivo therapeutic interventions indicated strong clinical relevance for our findings and highlighted ADAMTS4 as a promising therapeutic target in pulmonary fibrosis. Our study provides valuable insights into the cellular origins and trajectories of AFs and MyoFBs and sets the stage for identifying further targets for future interventions.

## Methods

### Animal experiments
All animal studies were conducted in accordance with the ARRIVE guidelines and in compliance with the approved protocols by the animal ethics committee of Justus-Liebig-University Giessen and the local authorities (Regierungspraesidium Giessen, permit numbers: G76/2020–No. 1026_GP and 825_M). All mice were kept under specific pathogen-free (SPF) conditions with unrestricted access to food and water. Housing was regulated at $22 \pm 2\,°C$ with $55 \pm 10\%$ relative humidity and a 14/10-h light/dark cycle. Eight-to-twelve-week-old wild-type (WT) mice with a C57BL/6 genetic background were obtained from Charles River (#632). Since aged female mice show better survival and less severe lung fibrosis in response to bleomycin instillation compared with aged male mice[32], they were chosen for the experiments. To lineage-label *Fgf10*+ cells, 52-to-56-week-old female *Fgf10*$^{Cre-ERT2/+}$; *tdTomato*$^{flox}$ mice were used. These mice were previously generated and validated in our lab[45].

### Human-derived lung tissues
Fresh human lung tissues from non-IPF donors used for precision-cut lung slice (PCLS) cultures were obtained from Hannover Medical School (MHH, Hannover, Germany) in compliance with 'The code of ethics of the world medical association' (Approval by the Ethics Committee of MHH renewed on 2015/04/22; approval number 2701-2015) and those from IPF patients from the UGMLC Giessen Biobank, member of the DZL Platform Biobanking (Approval by the Ethics Committee of Justus-Liebig-University Giessen; approval number AZ 58/15). Formalin-fixed, paraffin-embedded human donor and IPF lung sections used for in situ hybridization and fresh material used for RNA extraction followed by qPCR were obtained from the UGMLC Giessen Biobank (Approval by the Ethics Committee of Justus-Liebig-University Giessen; approval number AZ 58/15). The use of human samples was in accordance with the Declaration of Helsinki and written informed consent was obtained from all participants or their next of kin.

### Tamoxifen and bleomycin treatments
To label *Fgf10*+ cells, mice were intraperitoneally (i.p.) injected with tamoxifen (Sigma-Aldrich, T5648-5G) reconstituted in corn oil (Sigma-Aldrich, C8267) at a dose of 0.1 mg g$^{-1}$ body weight. Mice received three successive injections on days −13, −11 and −9 prior to intratracheal (*i.t.*) instillation of saline or bleomycin (1.5–2 U kg$^{-1}$ body weight; Bleomedac, PNZ-02411351). Lungs were collected on days 14, 30 and 60 *post* bleomycin injury for further analysis.

### Hematoxylin and eosin staining and fibrosis area quantification
Mouse lung perfusion was done using Hank's Balanced Salt Solution (HBSS; Thermo Fisher Scientific, 14175095). Once harvested, the lungs were fixed with 4% paraformaldehyde (PFA; Merck, 104005) and sequentially dehydrated through a series of ethanol concentrations. The lungs were then embedded in paraffin and cut into 5-μm-thick sections using a microtome (Leica, RM2255). Before staining, paraffin sections were deparaffinized with xylol and sequentially rehydrated. The slides were stained with hematoxylin (Roth, T865.2) for 2 min, followed by eosin (Thermo Fisher Scientific, 6766007) for 1 min. The stained slides were dehydrated and mounted. The damaged area was quantified by Orbit image analysis software (V3.64, Idorsia Pharmaceuticals Ltd).

### Picro Sirius Red and Masson's trichrome staining
Five-μm-thick paraffin sections were deparaffinized and rehydrated. For Picro Sirius Red staining (Abcam, ab150681), an adequate amount

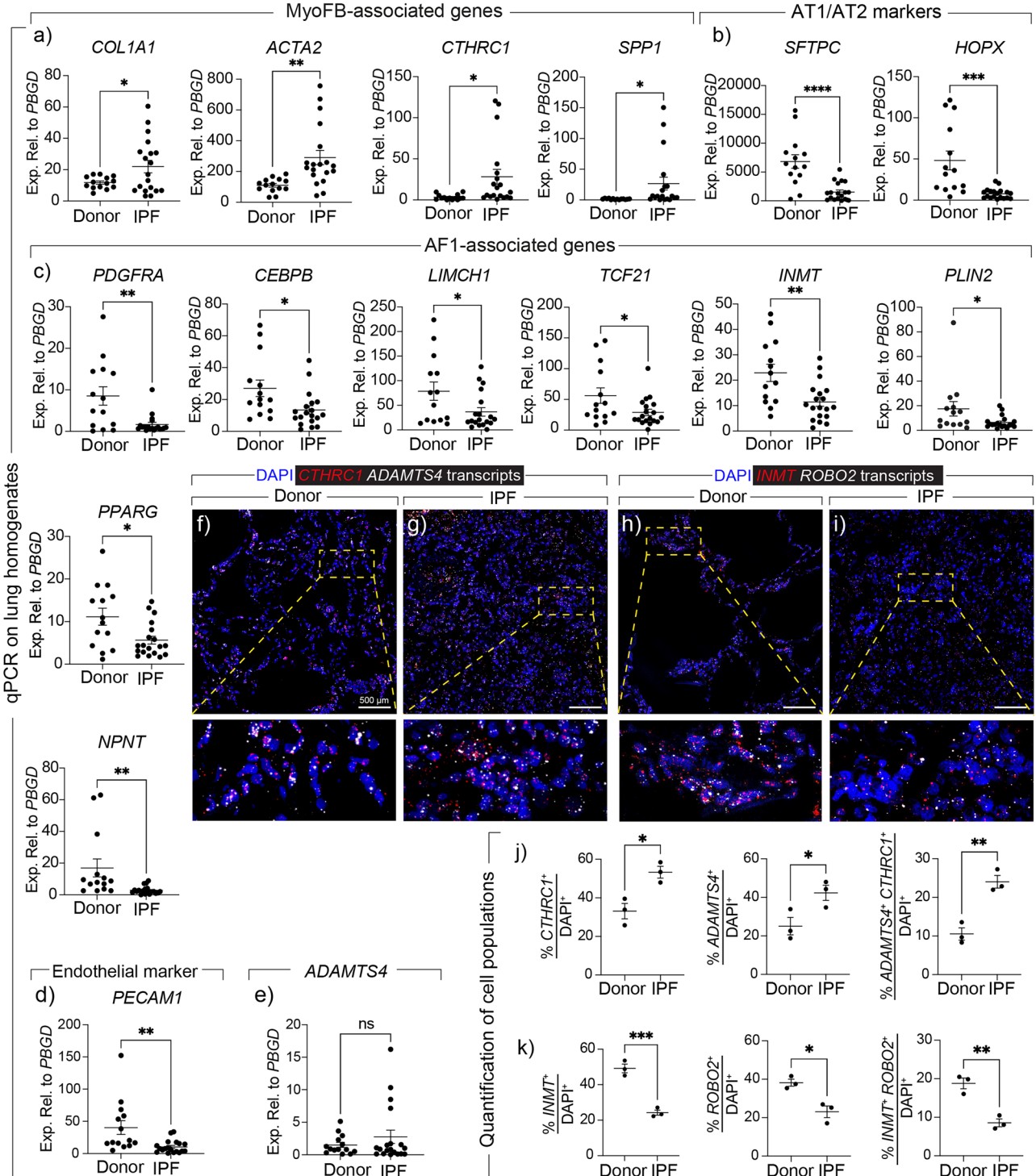

**Fig. 6 | Characterization of lipogenic and myogenic markers in human IPF lungs. a–e** Quantitative PCR for the indicated genes on lung homogenates from donor and IPF lung samples. **f, g** Representative images of in situ hybridization for *CTHRC1* (red) and *ADAMTS4* (white) in donor and IPF lung tissues. The dashed boxes are magnified in the lower panels. **h, i** Representative images of in situ hybridization for *INMT* (red) and *ROBO2* (white) in donor and IPF lung tissues. The dashed boxes are magnified in the lower panels. **j, k** Quantification of in situ hybridization data. Scale bars: 500 μm. **a–e** *n* = 14 for donors; *n* = 19 for IPF. **j, k** *n* = 3 per group. Each data point represents one patient. Data are presented as mean ± SEM. Statistical analysis was performed using student's *t* test (unpaired, two-tailed). *P < 0.05; **P < 0.01; ***P < 0.001; ****P < 0.0001; ns Not significant. *ACTA2* Actin

alpha 2, smooth muscle, *ADAMTS4* ADAM metallopeptidase with thrombospondin type 1 motif 4, *CEBPB* CCAAT enhancer binding protein beta, *CTHRC1* Collagen triple helix repeat containing 1, *COL1A1* Collagen type I alpha 1 chain, *HOPX* HOP homeobox, DAPI 4',6-diamidino-2-phenylindole, *INMT* Indolethylamine N-methyltransferase, IPF Idiopathic pulmonary fibrosis, *LIMCH1* LIM and calponin homology domains 1, *NPNT* Nephronectin, *PDGFRA* Platelet-derived growth factor receptor alpha, *PECAM1* Platelet and endothelial cell adhesion molecule 1, *PLIN2* Perilipin 2, *PPARG* Peroxisome proliferator activated receptor gamma, *SPP1* Secreted phosphoprotein 1, *TCF21* Transcription factor 21, *ROBO2* Roundabout guidance receptor 2, SFTPC Surfactant protein C.

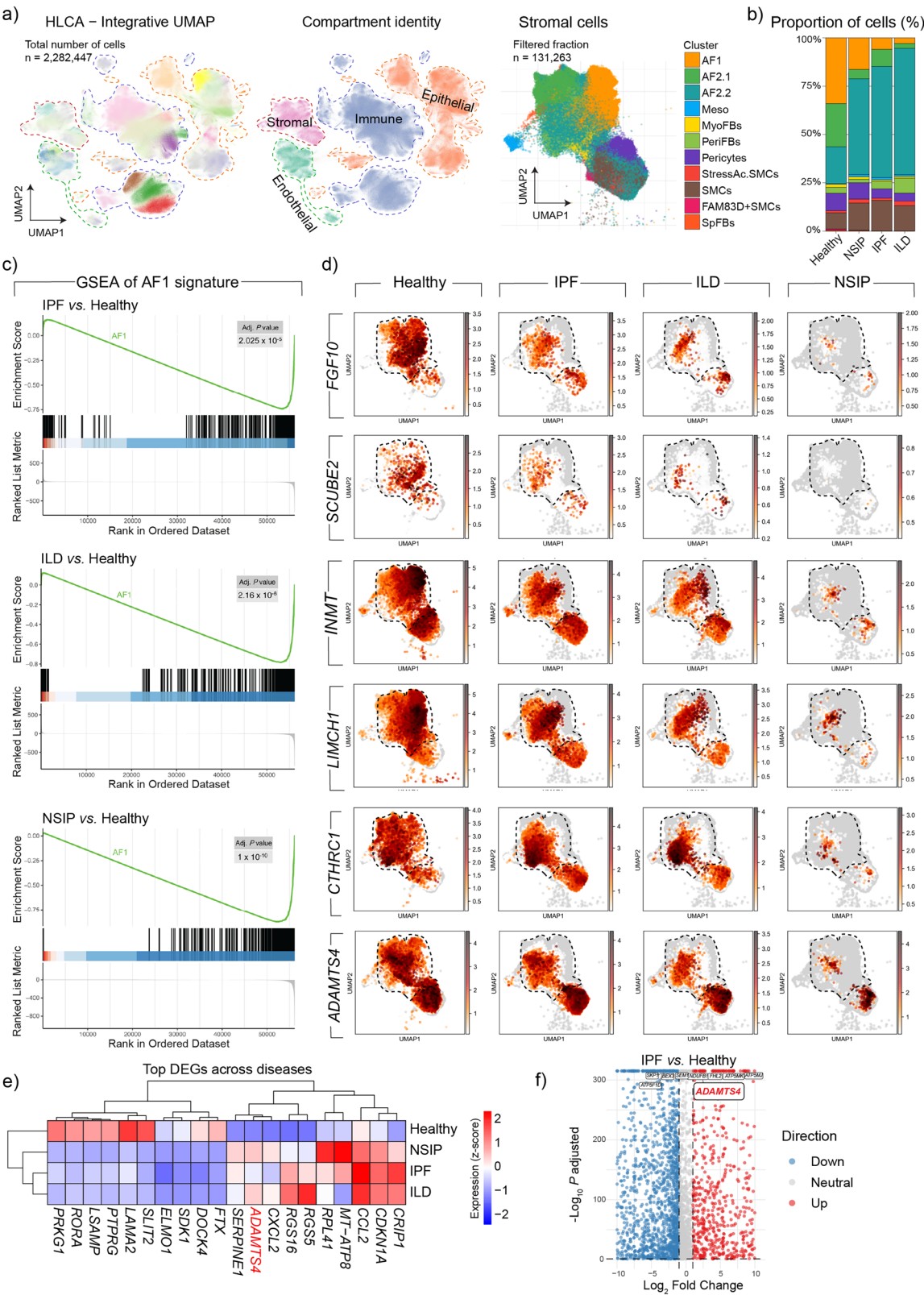

of dye was applied to cover the tissue and incubated for 1 h at room temperature (RT). The dye was washed in two changes of 1% acetic acid. Masson's Trichrome staining (Abcam, ab150686) was carried out according to the manufacturer's instructions. Stained sections were dehydrated, mounted and finally imaged.

### RNA extraction and qPCR

RNA was extracted using Trizol (Thermo Fisher Scientific, 10296028) and RNeasy mini kit (Qiagen, 74106). Complementary DNA (cDNA) was synthesized from total RNA using the QuantiTect reverse transcription kit (Qiagen, 205311) according to the manufacturer's instructions.

**Fig. 7 | Increased expression of *ADAMTS4* in fibrotic human lung disease.**
**a** Integrative UMAP of healthy, IPF, ILD and NSIP. **b** Frequency of clusters in each patient group. **c** GSEA of the *AF1* signature in IPF, ILD, and NSIP versus healthy datasets. Enrichment scores were calculated using the fgsea algorithm with genes ranked by log₂ fold change. *P*-values were adjusted for multiple comparisons using the Benjamini-Hochberg method, and exact adjusted *P* values (Adj. *P* value) are displayed for each plot. **d** UMAP plots for the indicated genes in healthy, IPF, ILD, and SNIP datasets. **e** Heatmap of the top DEGs across diseases. Expression values are represented as Z-scores of mean log-normalized counts. Differential expression was determined using a two-sided Wilcoxon Rank Sum test. **f** Volcano plot showing

top upregulated genes in IPF versus healthy. Statistical significance was calculated using a two-sided Wilcoxon Rank Sum test with Bonferroni correction for multiple comparisons. Only adjusted *P* values are reported. Source data are provided as a Source Data file. *ADAMTS4* ADAM metallopeptidase with thrombospondin type 1 motif 4, AF1 Alveolar fibroblasts 1, AF2 Alveolar fibroblasts 2, GSEA Gene set enrichment analysis, HLCA Human lung cell atlas, ILD Interstitial lung disease, IPF Idiopathic pulmonary fibrosis, Meso Mesothelial cells, MyoFB Myofibroblasts, NSIP Non-specific interstitial pneumonia, PeriFB Peribronchial fibroblasts, SMC Smooth muscle cells, SpFB Subpleural fibroblasts.

Quantitative real-time PCR (qPCR) was performed using PowerUp SYBR green master mix (Thermo Fisher Scientific, A25742) and a QuantStudio™ 3 Real-Time PCR System machine (Applied Biosystems™). Primer sequences are listed in Supplementary Data 1.

### ADAMTS4 activity assay

To isolate protein, human or murine PCLSs were lysed using 150 μL of RIPA buffer (50 mM Tris-HCl pH 7.5, 0.15 M NaCl, 1% NP40). Briefly, PCLSs were cut into small pieces using scissors and incubated in 4 °C for 30 min. Then, the lysates were centrifuged at $10,000 \times g$ for 10 min at 4 °C. Next, the supernatant was isolated and used for protein measurement with DC Protein Assay Reagents Package (BioRad) according to the manufacture's protocol. ADAMTS4 activity was determined using the ADAMTS4 Fluorogenic Assay Kit (BPS Bioscience, 82546) according to the manufacturer's instructions. Measurements were acquired using an Infinite M200 plate reader (Tecan) equipped with i-control 2.0 software (Tecan). Activity was calculated as follows to yield fluorescence units/μg:

$$\frac{\text{Sample reading} - \text{Blank reading}}{\text{Protein concentration}} * 1000$$

### Fluorescence-activated cell sorting

Mice were euthanized and lungs were perfused with 20 mL HBSS through the right ventricle. To isolate tdTom+ cells for single-cell RNA sequencing (scRNA-seq), lungs from *Fgf10^Cre-ERT2/+; tdTomato^flox* mice were chopped into small pieces and digested using 0.5% collagenase type IV (Gibco, 17104-019) in HBSS for 30 min on a rotator at 37 °C. After digestion, lung homogenates were passed through 70 μm and 40 μm cell strainers (Sarstedt, 83.3945070 and 83.3945.040, respectively) to obtain single-cell suspensions. After centrifugation, the pellet was resuspended in an antibody cocktail containing anti-mouse CD31 (AF488-conjugated, Biolegend, 102514, 1:100), CD45 (AF488-conjugated, Biolegend, 103121, 1:100) and CD326 (EpCAM, APC/Cy7-conjugated, Biolegend, 118218, 1:50) for 20 min on ice in the dark. After washing, cells were stained with SyTOX (1:1000, Invitrogen, S34862) to exclude dead cells and sort live cells for scRNA-seq.

For murine alveolosphere assays, freshly isolated resident mesenchymal cells (rMCs) identified as CD45^neg CD31^neg EpCAM^neg Sca-1^pos cells were co-cultured with freshly isolated alveolar epithelial cells (AECs) as previously described[53]. In brief, lungs from young WT mice were inflated i.t. with 3 mL dispase (5 U/mL; BD Biosciences, 354235) and further digested by incubation with 2 mL dispase solution (5 U/mL) for 40 min at RT. The cell suspension was incubated with biotin rat anti-mouse CD45 (BD Biosciences, 553078), CD16/32 (BD Biosciences, 553143), and CD31 (BD Biosciences, 553371) for 30 min to deplete hematopoietic and endothelial cells, respectively, using magnetic separation. The cell pellet was incubated in the antibody cocktail (anti-CD31, anti-CD45 and anti-CD326) containing gamma globulin (sandoglobulin) to sort CD45^neg CD31^neg EpCAM^low AECs. Anti-mouse Ly-6A/E (Pacific Blue-conjugated, Biolegend, 108120, 1:50) antibodies were used to isolate rMCs. FACSAria III cell sorter (BD Biosciences) was used

for sorting and measurements. At least 100,000 rMCs and 400,000 AECs from one WT lung could be sorted. Data were analysed using FlowJo software (FlowJo LLC, version 10.10.0).

### Alveolosphere cultures

The alveolosphere assay was performed as previously described[35]. In brief, 17,000 FACS-sorted rMCs and 5000 AECs from young WT mice were resuspended in 50 μL organoid medium containing minimum essential medium (MEM) Alpha (Gibco, 41061029), 10% FBS, 1% L-glutamine-penicillin-streptomycin (Sigma-Aldrich, G1146), 1% insulin/transferrin/selenium (ITS) (Gibco, 41400-045) and 0.0002% heparin (Stem cell Technologies, 07980). The cells were mixed at a ratio of 1:1 with cold growth factor-reduced phenol red-free Matrigel (Corning, 356231), transferred to a 24-well plate on a 0.4 μm cell culture insert (Millipore, PICM01250) and incubated for 15 min for Matrigel polymerization. Next, 350 μL of organoid medium was added to the lower chamber. Cultures were incubated under air-liquid conditions at a temperature of 37 °C with 5% $CO_2$ for three weeks. The culture medium was changed every other day. Alveolospheres were imaged using EVOS M7000 (Thermo Fisher Scientific, AMF7000). The formed WT alveolospheres were treated with 5 ng/mL rTGFβ1 for three days, followed by rTIMP-3 (300 ng/mL) treatment for another three days. The alveolospheres were used for qPCR and IF. Primer sequences are listed in Supplementary Data 1. RNA extraction was done using micro kit (Qiagen, 74004).

### Whole-mount immunofluorescence of alveolospheres

Cultured alveolospheres were fixed with 4% PFA for 20 min at RT and washed with 1X PBS. Samples were blocked and permeabilized with 0.5% Triton-X-100 and 5% donkey serum in 1X PBS overnight at 4 °C. Primary anti-Pro-SFTPC and mouse/rat anti-Rage (R&D Systems, MAB1179, 1:200) antibodies were diluted in the blocking buffer and alveolospheres were incubated overnight at 4 °C. Samples were washed three times using washing buffer (2% donkey serum, 0.3% Triton-X-100 in 1X PBS) followed by incubation with the corresponding secondary antibodies. Nuclei were subsequently counterstained with DAPI. The samples were covered with cover slips and ProLong™ Glass Antifade Mountant (Thermo Fisher Scientific, P36984).

### In situ hybridization, immunofluorescence and quantification

Five-μm-thick paraffin sections from mice or human transplanted lungs were deparaffinized and rehydrated. RNAscope was performed according to the manufacturer's instructions (ACD, Doc. No. 323100-USM) using mouse probes for *Fgf10* (Mm-Fgf10, 446371-C1), *Inmt* (Mm-Inmt, 486371-C1), *Robo2* (Mm-Robo2, 475961-C2), *Cthrc1* (Mm-Cthrc1, 413341-C1) and *Adamts4* (Mm-Adamts4, 537341-C2) or human probes for *CTHRC1* (Hs-Cthrc1, 413331-C1), *ADAMTS4* (Hs-ADAMTS4, 497161-C3), *INMT* (Hs-INMT, 459961-C1) and *ROBO2* (Hs-ROBO2, 502341-C2). RNAscope for *Tcf21* (Mm-Tcf21, 508661-C1) was followed by immunofluorescence (IF) for SFTPC (Seven Hills Bioreagents, WRAB-9337, 1:1000), ACTA2 (Sigma, F3777-.2 ML, 1:100), or ADAMTS4 (Thermo Fisher Scientific, PA1-1749A, 1:150). Anti-tdTomato antibody (Sicgen, AB8181-200, 1:250) was used to boost the tdTomato fluorescent signal

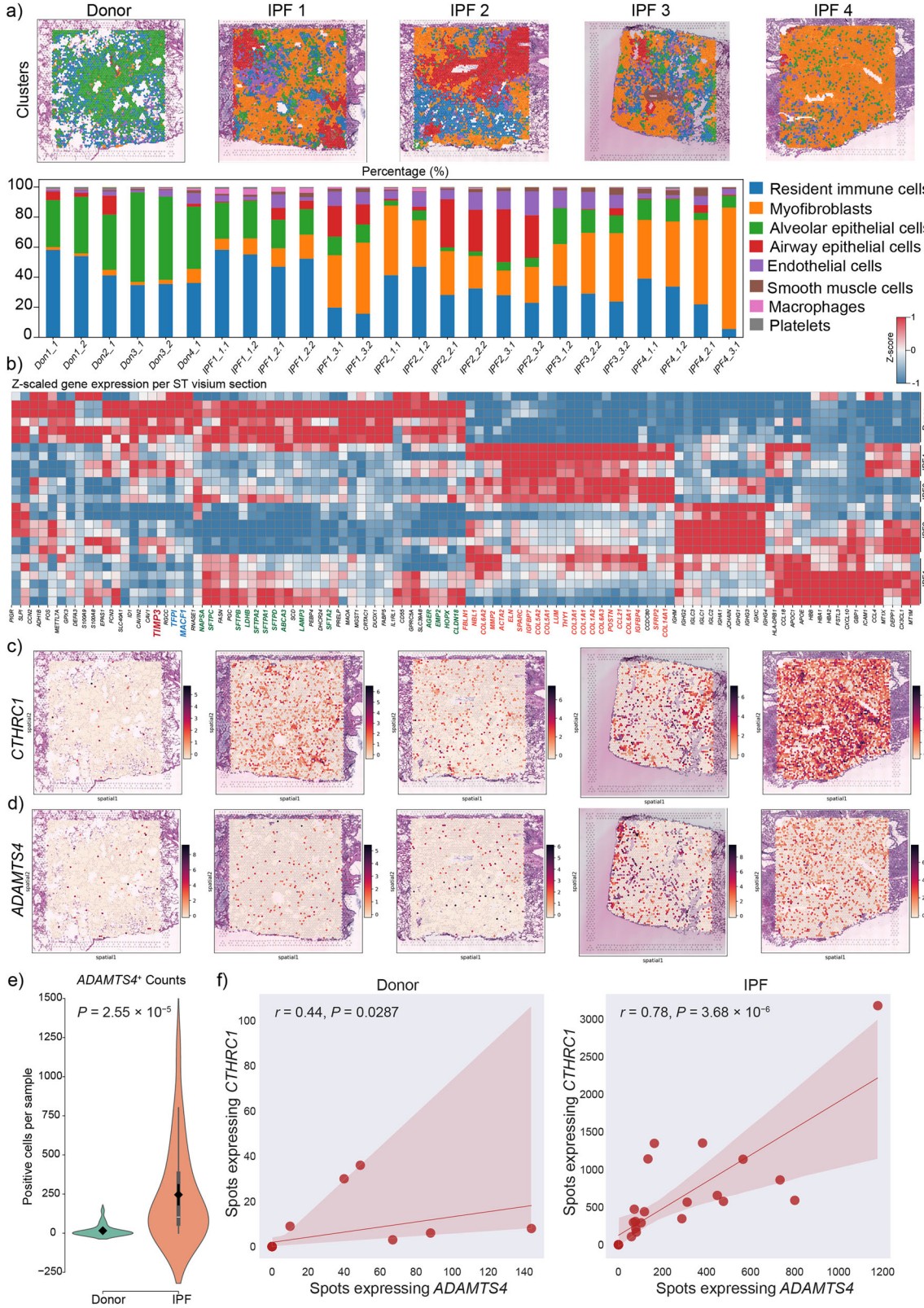

that was diminished after antigen retrieval. For IF, the samples were blocked with 5% BSA serum (Jackson ImmunoResearch, 001-000-162) for 1 h at RT, followed by incubation with primary antibodies overnight at 4 °C. The next day, samples were washed three times with PBS, then incubated with secondary antibodies for 1 h at RT. Nuclei were counterstained using 4´,6-diamidino-2-phenylindole (DAPI; Thermo Fisher Scientific, D1306) and slides were mounted by Fluoromount-G (Southern Biotech, 0100-01). Fluorescent images were acquired using Mica Widefocal LiveCell System (Leica microsystem). Images were quantified using Fiji software[54].

**Fig. 8 | Spatial transcriptomics reveals upregulation of *ADAMTS4* in human IPF.**
**a** Spatial plots of lung tissue sections from donor and IPF samples with cellular composition and organization. **b** Heatmap of Z-scaled expression for top differentially expressed genes (50 per condition) across spatial transcriptomic samples. Gene expression was averaged per sample, normalized to z-scores and hierarchically clustered using average linkage. Rows represent samples ordered by condition; columns represent genes sorted by clustering. **c, d** Spatial feature plots of *CTHRC1* and *ADAMTS4*. **e** Violin plot comparing the number of spots expressing *ADAMTS4* between donor and IPF samples. Data are presented as mean values +/- SE. Internal box plots indicate the median and interquartile range, with whiskers representing the minimum and maximum. Statistical significance was assessed using a two-sided Mann–Whitney U test. **f** Correlation analysis between spots expressing *CTHRC1* and *ADAMTS4* in donor and IPF. Pearson correlation coefficients (*r*) and exact *P* values were calculated using a two-sided Pearson correlation test. The regression line includes a 95% confidence interval represented by the shaded error band. Source data are provided as a Source Data file. Statistical analysis was performed using Mann–Whitney U test for group comparisons and Pearson correlation for associations. *ADAMTS4* ADAM metallopeptidase with thrombospondin type 1 motif 4, *CTHRC1* Collagen triple helix repeat containing 1, Donor healthy lung, IPF idiopathic pulmonary fibrosis.

## Generation of human and mouse precision-cut lung slices (PCLS) and treatments

Fresh human lung tissues from IPF patients that underwent lung transplantation were inflated with 2% UltraPure low melting agarose (Invitrogen, 16520-100) and cut into 300 μm-thick slices using a vibratome. The resulting human precision-cut lung slices (hPCLS) were cultured in DMEM/F-12, GlutaMAX™ medium (Thermo Fisher Scientific, 31331028) supplemented with 10% fetal bovin serum (FBS) (Gibco, 10270-106), 1% penicillin-streptomycin (Thermo Fisher Scientific, 15070063) and 0.5% amphotericin (Sigma, A2942). Cultures were treated with vehicle, 300 ng/mL recombinant protein TIMP-3 (rTIMP-3) (R&D Systems, 973-TM-010) or 500 ng/mL recombinant versican (rVCAN) (abbexa, abx069660) in medium containing 0.1% serum for three days. Mouse PCLS were generated as described[55] at a thickness of 200 μm and treated with vehicle or 5 ng/mL rTGFβ1 (R&D Systems, 240-B) for five days followed by treatment with vehicle, rTGFβ1 + Vehicle or rTGFβ1 + rTIMP-3 for three days. PCLS obtained from Bleo d14 *Fgf10^{Cre-ERT2}; tdTomato^{flox}* mice were treated with vehicle or 500 ng/mL rVCAN for three days. The samples were collected for RNA extraction and qPCR.

Donor non-IPF PCLS were prepared from healthy margins of surgically resected lungs of patients with lung cancer. Briefly, human lung lobes were cannulated with a flexible catheter and the selected lung segments were inflated with warm (37 °C) low melting agarose (1.5%) prepared in Dulbecco's Modified Eagle's Medium Nutrient Mixture F-12 Ham (DMEM-F-12) supplemented with 15 mM HEPES, 100 U/mL penicillin, and 100 μg/mL streptomycin (all from Invitrogen Life Technologies). After polymerization of the agarose solution on ice, tissue cores of a diameter of 8 mm were prepared using a sharp rotating metal tube. Subsequently, the cores were sliced into 250–300 μm-thin slices in DMEM using a Krumdieck tissue slicer (Alabama Research and Development). PCLS were washed three times for 30 min in DMEM-F-12 supplemented with 15 mM HEPES, 100 U/mL penicillin, and 100 μg/mL streptomycin and used for the experiments.

## *ADAMTS4* gene silencing in PCLS

PCLS generated from human donor lungs were placed in 12-well-plates in 800 μL DMEM F-12 medium supplemented with HEPES and penicillin/streptomycin. Afterward, PCLS were transfected with lipoplexes of siLentFect lipid reagent (Bio-Rad Laboratories) encapsulating 100 nmol pre-designed, commercially available siRNA against *ADAMST4* (Thermo Fisher Scientific, AM16830) in 200 μL OPTI-MEM. To control for non-specific gene inhibition of the siRNA used in this study, a universal negative-control scrambled siRNA (si-Scr) sequence was employed (Thermo Fisher Scientific, 4390843). The tissue slices were incubated for 48–72 h at 37 °C and 5% CO$_2$ and afterward collected for fixation (followed by immunofluorescence and in situ hybridization), RNA extraction (for qPCR), or protein isolation (for activity assays or western blots). Western blots were performed according to standard procedures using antibodies against ADAMTS4 (Thermo Fisher Scientific, PA1-1749A; 1:1000), COL1 (Southern Biotech, 1310-1; 1:500), and Beta-actin (ACTB; Sigma, A1978; 1:5000). The latter was used as a loading control. Secondary anti-rabbit (Dako, P0217; 1:2000), anti-goat (Dako, P0449; 1:2000), and anti-mouse (Dako, P0447; 1:5000) antibodies were used, respectively. Blots were developed using SuperSignal West Dura Extended Duration Substrate (Thermo Fisher Scientific, 34075) and imaged using ChemiDoc Touch Imaging System and Image Lab 6.1 software (Bio-Rad).

## Single-cell RNA sequencing

To explore transcriptional changes specific to mesenchymal cells in the aged *Fgf10^{Cre-ERT2}; tdTomato^{flox}* animals, scRNA-seq was conducted on tdTom+ cells extracted from the lungs of saline or bleomycin-challenged mice. Cells were flow-sorted, and libraries were prepared using the Chromium Next GEM Single-Cell 3′ Reagent Kit v3.1 from 10x Genomics. Sequencing data underwent processing through a previously established pipeline[56], which in brief involved excluding cells with fewer than 1,200 detected counts or more than 40,000 counts, as well as those with over 20% mitochondrial gene expression. Genes detected in fewer than 20 cells were filtered out to ensure high-quality read alignment and gene quantification. Following the pre-processing of individual samples, data integration was performed using the Harmony algorithm, with batch correction applied through BBKNN (Batch Balanced K-Nearest Neighbors) and highly variable genes selected using SeuratV3 flavor.

Differentially expressed genes (DEGs) were identified using the presto package in Seurat, with a minimum log-fold change threshold of 0.5 and a detection rate of at least 50%. To compare differences between clusters and samples, differential expression analysis was performed using the *FindMarkers* function, followed by filtering and visualization of significant genes through a volcano plot using tidyplots[57] or with the pheatmap package. RNA velocity analysis was conducted to map transcriptional dynamics and cell state predictions were explored using scVelo package, selecting 500 highly variable genes and computing first- and second-order moments with 20 principal components and 30 nearest neighbors. Latent cell velocity and inferred differentiation trajectories were further inferred using CellRank[58], employing standard settings and vignette, enabling probabilistic modeling of cell-fate transitions. Cluster activity along latent time was quantified by mapping cluster identities onto the trajectory and visualizing smoothed density curves, highlighting early, transitional, and late contributors to the dynamic process. To gain functional insights, cluster markers were mapped to ENTREZ gene identifiers and annotated using the org.Mm.eg.db database. Pathway enrichment analyses were conducted using Gene Ontology (GO) and Kyoto Encyclopaedia of Genes and Genomes (KEGG) databases to identify enriched biological processes and signaling pathways, with results visualized using ggplot2 and dplyr.

Stromal cells from healthy, NSIP, IPF, and ILD donor lungs were obtained from the Human Lung Cell Atlas (HLCA)[59]. Differential expression analysis was conducted by comparing IPF, NSIP and ILD to healthy stromal cells. DEGs were calculated by comparing each disease category against 'normal' (healthy stromal cells) using the Wilcoxon rank-sum test. Ensembl IDs mapped to the gene symbols were incorporated into the final data frame, and genes with infinite avg_log2FC values were excluded. Key transcriptional changes associated with the

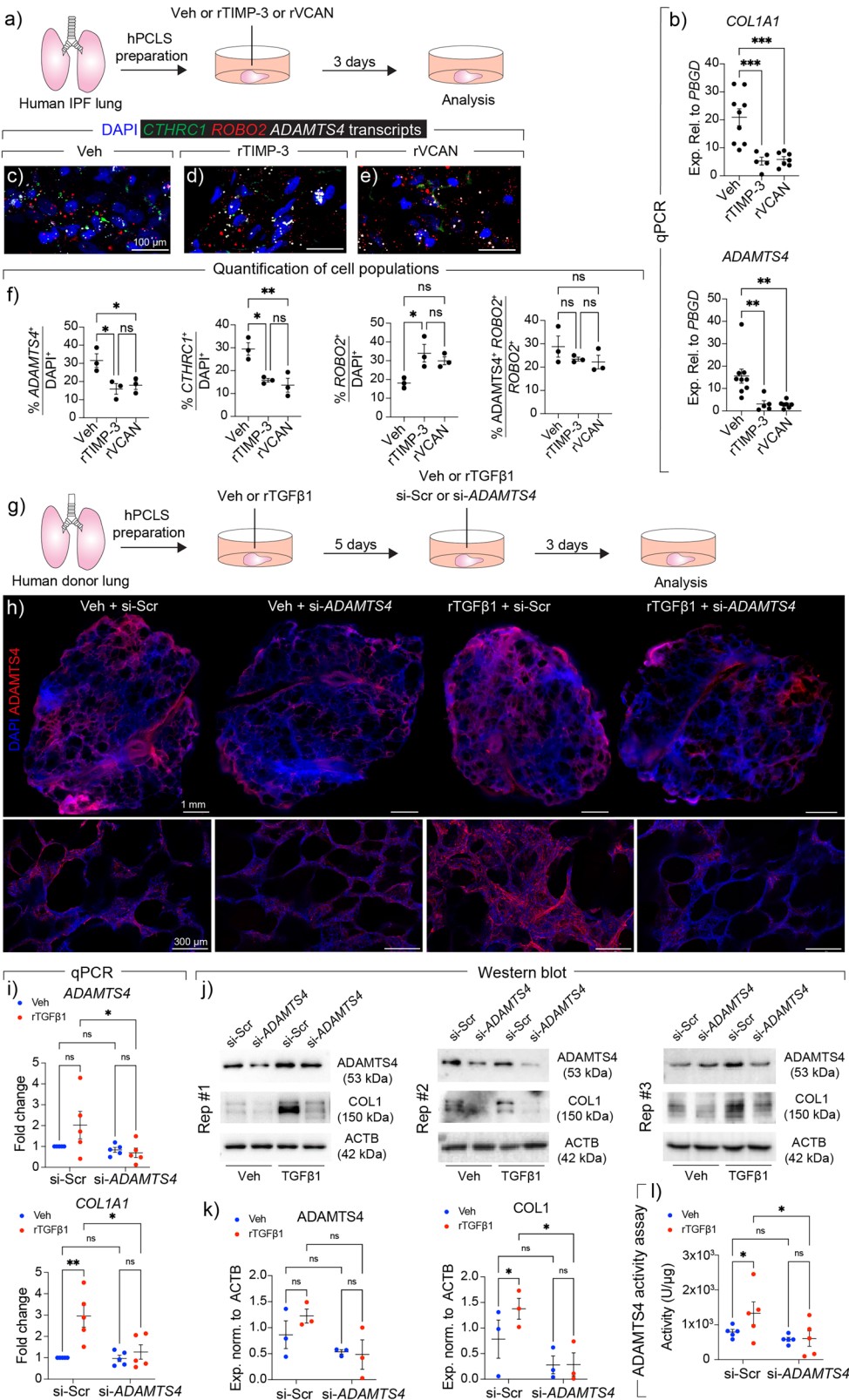

AF1 signature (Source Data file) were analysed using the clusterProfiler and fgsea packages.

Spatial transcriptomic analyses were performed using Squidpy[60]. Visium libraries were subsetted while retaining spatial coordinates, and gene expression matrices were normalized prior to non-negative matrix factorization (NMF) to identify spatially structured transcriptional programs. NMF factor assignments were stored in the observation table and visualized with Squidpy, and factor loadings were compared to assess correspondence between spatial programs. Spot-level gene detection frequencies were quantified across conditions, group differences evaluated with Mann–Whitney U tests, and Pearson correlations of gene-positive spots computed to examine co-expression within each timepoint. All Visium datasets, including Space Ranger outputs

**Fig. 9 | *ADAMTS4* loss of function attenuates fibrogenesis in human precision-cut lung slice cultures. a** Timeline and schematic of the experimental design. **b** Quantitative PCR for the indicated genes. **c–e** Representative images of in situ hybridization for *CTHRC1* (green), *ROBO2* (red), and *ADAMTS4* (white) in different groups. **f** Quantification of in situ hybridization data. **g** Timeline and schematic of the experimental design. **h** Representative tile scans of immunofluorescence for ADAMTS4 in hPCLS from different experimental groups. High-magnification images are shown in the lower panels. **i** Quantitative PCR for the indicated genes. Hypoxanthine guanine phosphoribosyl transferase (*HPRT*) was used as a reference gene. **j** Western blots for the indicated proteins showing three biological replicates (three patients). For technical reasons, the COL1 blot for Replicate #1 was run on a separate gel. **k** Densitometry analysis of western blots. **l** ADAMTS4 activity assay. Source data are provided as a Source Data file. Scale bars: **c–e** 100 μm; **h** 1 mm

(upper panel) and 300 μm (lower panel). **b** *n* = 9 for Veh; *n* = 5 for rTIMP-3; *n* = 7 for rVCAN; **f** *n* = 3 per group; **i** *n* = 5 per group; **k** *n* = 3 per group; **l** *n* = 5 per group. Each data point represents one well in (**b**, **f**) and one patient in (**i–l**). Data are presented as mean ± SEM. Statistical analysis was performed using ordinary one-way ANOVA with Tukey's multiple comparisons test for (**b**, **f**) and RM two-way ANOVA with multiple comparisons for (**i**, **k**, **l**). *\*P* < 0.05; *\*\*P* < 0.01; *\*\*\*P* < 0.001; ns Not significant. *ACTB* Beta-actin, *ADAMTS4* ADAM metallopeptidase with thrombospondin type 1 motif 4, *COL1* Collagen type I, *COL1A1* Collagen type I alpha 1 chain, *DAPI* 4′,6-diamidino-2-phenylindole, *hPCLS* Human precision-cut lung slices, *IPF* Idiopathic pulmonary fibrosis, *ROBO2* Roundabout guidance receptor 2, *Rep #* Replicate number, *rTGFβ1* recombinant transforming growth factor beta 1, *rTIMP-3* recombinant tissue inhibitor of metalloproteinases, *rVCAN* Recombinant versican, *si-ADAMTS4* siRNA targeting *ADAMTS4*, *si-Scr* Scrambled siRNA, *Veh* Vehicle.

and full-resolution images, were obtained from BioStudies (S-BSST1410, human)[38].

## Statistical analysis and figure preparation
All data are presented as mean ± standard error of the mean (SEM). For comparison between two groups, student's *t* test (unpaired, two-tailed) was employed. One-way ANOVA was used for comparisons among three or more groups with a single variable and two-way ANOVA was used for comparisons with two variables. Statistical tests are indicated in the corresponding figure legends. The *P* < 0.05 was considered statistically significant. Quantitative data were assembled and analyzed using GraphPad Prism (GraphPad Software, version 10.4.1 (532)). The number of biological replicates (depicted as n) is indicated in the corresponding figure legends. Figures were prepared and assembled using Adobe Illustrator (Adobe).

## Reporting summary
Further information on research design is available in the Nature Portfolio Reporting Summary

## Data availability
Cluster quantification data, DEGs, gene signatures used for GSEA, and scoring are provided in the Source Data file. Raw and processed data were deposited on Gene Expression Omnibus (GEO) and are publicly available under the accession number GSE295566. Source data are provided with this paper.

## Code availability
The analysis code is available through: https://github.com/alikhadim/ADAMTS4-Stromal-Remodeling/.

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

## Acknowledgements

The authors thank Martin R Stampfli and Giuseppina Fascellaro for their critical input. We also acknowledge David Dippel for his technical assistance with the human precision-cut lung slice cultures and siRNA experiments, Kerstin Goth and Hannah Hofmann for the animal husbandry and genotyping, and Thomas Sontag and Ingrid Henneke for their help with the animal protocols. This study was supported by funding from CSL through their Research Acceleration Initiative (https://www.csl.com/csl-rai). A.L. was supported by an Institute for Lung Health (ILH) startup grant and the Excellence Cluster Cardio-Pulmonary System/Cardio-Pulmonary Institute (EXC 2026, project number 390649896). S.H. was supported by German Research Foundation (DFG) grants SFB-TR84 (project number 114933180, project B2), SFB1021 (project number 197785619, project C05), and KFO309 (project number 284237345, projects P2, P8); the Excellence Cluster Cardio-Pulmonary System/Cardio-Pulmonary Institute (EXC 2026, project number 390649896); the German Center for Lung Research (DZL, project number 82DZL005B1), the Hessen State Ministry of Higher Education, Research and the Arts (HMWK, Landes-Offensive zur Entwicklung Wissenschaftlich-ökonomischer Exzellenz, LOEWE, Förderlinie 4a project ID III L7–519/05.00.002). E.E.A. was funded by the Institute for Lung Health (ILH), DFG (EL 931/4-2, EL 931/5-1, EL 931/4-1, EL 931/2-2 (KFO309 284237345), and SFB 1213 Project-ID 268555672), the Excellence Cluster Cardio-Pulmonary System/Cardio-Pulmonary Institute (EXC 2026, project number 390649896), and DZL.

## Author contributions

Conceptualization, E.E.A.; Investigation, M.Z., A.K., A.L., M.Ba., S.B. and E.E.A.; Methodology, M.Z., A.K., A.L., A.I.V-A., St.H., G.D.P., D.K., J.H., T.P.K., M.Ba., X.C., J.K., N.W., M.W., S.B. and E.E.A.; Writing the animal protocol, A.L.; Resources, C.S., M.Bo., N.W., A.G., W.S., P.B., S.H., M.W.,

S.B. and E.E.A.; Writing—Original Draft, M.Z., A.K., A.L. and E.E.A.; Writing—Review & Editing, A.K. and E.E.A.; Funding Acquisition, E.E.A.; Supervision, S.B. and E.E.A.

## Funding

## Competing interests
The authors declare no competing interests.

## Additional information

[1]Department of Medicine V, Internal Medicine, Infectious Diseases and Infection Control, Universities of Giessen and Marburg Lung Center (UGMLC), German Center for Lung Research (DZL), German Center for Infection Research (DZIF), Justus-Liebig-University Giessen (JLU), Giessen, Germany. [2]Cardio-Pulmonary Institute (CPI), Giessen, Germany. [3]Institute for Lung Health (ILH), Giessen, Germany. [4]Department of Medicine II, Internal Medicine, Pulmonary and Critical Care, Universities of Giessen and Marburg Lung Center (UGMLC), German Center for Lung Research (DZL), Justus-Liebig-University Giessen, Giessen, Germany. [5]Transdisciplinary Research Area Life and Health, Organoid Biology, Life & Medical Sciences Institute, University of Bonn, Bonn, Germany. [6]CSL Innovation GmbH, Marburg, Germany. [7]Biomedical Informatics and Systems Medicine, Justus-Liebig-University Giessen (JLU), Giessen, Germany. [8]Oujiang Laboratory (Zhejiang Lab for Regenerative Medicine, Vision and Brain Health), School of Pharmaceutical Science, Wenzhou Medical University, Wenzhou, Zhejiang, China. [9]School of Pharmaceutical Sciences, Wenzhou Medical University, Wenzhou, Zhejiang, China. [10]Institute for Pathology, Hannover Medical School, Hannover, Germany. [11]Department of Pulmonary and Critical Care Medicine, The Quzhou Affiliated Hospital of Wenzhou Medical University, Quzhou People's Hospital, Quzhou, Zhejiang, China. [12]Laboratory of Extracellular Matrix and Regeneration, Justus-Liebig-University Giessen (JLU), Cardio-Pulmonary Institute (CPI), German Center for Lung Research (DZL), Institute for Lung Health (ILH), Giessen, Germany. [13]These authors contributed equally: Mahsa Zabihi, Ali Khadim. ✉e-mail: arun.lingampally@innere.med.uni-giessen.de; saverio.bellusci@innere.med.uni-giessen.de; elie.el-agha@innere.med.uni-giessen.de

