## [Transparent Peer Review file · Nature Communications]

Persistence of alveolar fibroblast-derived ADAMTS4+ cells in a preclinical model of delayed pulmonary fibrosis resolution

Corresponding Author: Professor Elie El Agha

Version 0:

Reviewer comments:

Reviewer #1

(Remarks to the Author)

In their manuscript, "Persistence of alveolar fibroblast-derived ADAMTS4+ cells during delayed resolution of pulmonary fibrosis," Zabihi et al. present an in-depth transcriptomic and tissue imaging analysis of lung mesenchymal states associated with delayed resolution of pulmonary fibrosis in mouse models and with idiopathic pulmonary fibrosis in humans. The authors use an aged mouse model to investigate how Fgf10-lineage traced lung mesenchymal cells determine the kinetics of resolution of pulmonary fibrosis following bleomycin-induced injury. Building on their previous work, the authors found that some of the Fgf10+ mesenchymal cells have a lipofibroblast/AT1 signature and these cells acquire markers associated with myofibroblasts. During resolution of fibrosis, these lineage-traced cells can interconvert back to a state with a lipofibroblast/AT1 signature. Analysis of the dynamics of Fgf10-lineage traced cells during development and resolution of fibrosis in mouse models found that expression of the gene *Adamts4* was enriched in areas of non-resolving fibrosis. The authors then performed in vitro experiments in lung organoids and ex vivo experiments in precision cut lung slices to test the therapeutic effect of targeting ADAMTS4 activity and one of its known extracellular matrix substrates, versican. To determine the relevance of these findings to human lung health, the authors analyzed lung tissue from IPF patients and precision cut lung slices from humans. They found that lung cells expressing ADAMTS4 and another marker of fibroblasts associated with pulmonary fibrosis, CTHCR1, were enriched in tissue from IPF patients compared to controls.

The study provides novel insight into mesenchymal cell states and fibroblast-produced molecules that may prevent resolution of fibrosis following lung injury. The experiments are well designed and the methodology is sound. However, additional analyses are required to determine the fibroblast lineages that give rise to the ADAMTS4+ fibroblasts associated with delayed resolution. Additional experiments are also needed to mechanistically link ADAMTS4 activity to the fibrogenesis observed in in vitro and ex vivo models. Addressing the following major and minor comments would provide further support for the authors' conclusions and improve the manuscript.

Major Comments:

1. In Figure 1, the authors provide convincing evidence that Fgf10 lineage-traced cells increase in frequency following bleomycin-induced lung injury and that expression of markers associated with a lipofibroblast phenotype (*Tcf21*, *Inmt*, and *Robo2*) decreases among these cells before increasing during injury resolution. In line 250, the authors state that the Fgf10-lineage traced "tdTom+ cells were massively recruited to the fibrotic regions..."

Do the authors have any additional evidence that the tdTom+ cells are recruited to the fibrotic regions from other sites as opposed to expanding as a result of local proliferation of these cells?

In order to address this, the authors could assess proliferation in the tdTom+ cells or clarify what they mean by "recruitment."

2. Related to the comment above, the authors show in Figure 2 that Fgf10 lineage-traced cells, as identified by tdTomato expression, represent a heterogeneous population of mesenchymal cells, including AF1, AF2, and peribronchial fibroblasts. These results appear to be in contrast to the authors statement in the introduction (lines 104 - 107) that lipofibroblasts are characterized by expression of Fgf10.

Is the heterogeneity observed in the scRNAseq data a result of an impure population of tdTomato+ due to sorting? Or does

the scRNAseq data likely represent the true heterogeneity of Fgf10 lineage-traced cells?

In order to address this question, the authors could perform an analysis of their imaging data from the saline condition to determine if the tdTomato+ cells are located in regions of the lung where AF2 and peribronchial fibroblasts would be expected to be observed.

3. Figure 3b includes a heat map representing the differentially expressed genes between AF1 and MyoFB cells across different time points after bleomycin treatment. More details of this analysis are required to properly interpret the data.

What does the 'Expression level' represent in this analysis? Is it a pairwise comparison between the AF1 cluster and the MyoFB cluster for each of the genes at the indicated time point?

If so, I would have expected that markers gene of AF1 cells (and lipofibroblasts), including Scube2, Wnt2, and Npnt, to be enriched in the AF1 cells in the saline condition and not following bleomycin injury at day 14.

The authors could take advantage of module scoring as they did in Figure 2 h, i, o, and p to assess the dynamics of these different cell states across the time points.

4. In Figure 5, the authors perform elegant studies modeling aspects of fibrogenesis in mouse lung organoids and precision cut lung slices. They induce gene expression of markers of fibrosis using TGFB and are able to reverse this gene expression by treating the organoids with recombinant TIMP3 or the precision cut lung slices with versican. While both TIMP3 and versican interact with ADAMTS4, there is no direct evidence presented indicating that ADAMTS4 is present and active in these models.

Do the authors have any evidence that Adamts4 is expressed following TGFB treatment or that ADAMTS4 protein is produced and inhibited by TIMP3? Does knocking out or knocking down Adamts4 gene expression have the same effect as TIMP3 treatment?

Relatedly, in Figures 5 n - o, how do the authors propose that versican is having its effect in reducing expression of these markers of fibrosis? Is the versican directly affecting mesenchymal cell activation or does it interact with ADAMTS4 in a way that it saturates its protease activity?

Minor comments:

5. In Figure 2, there is an overwhelming number of analyses presented. Some of these panels could be moved to a supplemental figure. In addition, it would be useful to present the entire gating strategy used to sort tdTomato+ including the forward and side scatter getting along with any gating on a live/dead dye.

6. In Figure 3o, the authors report that ~25% of total cells (among DAPI+) are Cthrc1+ in the saline controls as measured by RNA in-situ hybridization. Can the authors comment on whether such a high frequency of cells expressing Cthrc1 was expected prior to bleomycin injury?

7. The authors use an analysis of 'inferred latent time' in several figures in the manuscript (4a, S2a). For these analyses, what does the color-coding indicate?

8. There is a minor typo in line 268 - change "ADATMS4" to "ADAMTS4"

9. For the quantification of the immunofluorescence imaging in Figures 1, 3, 4, and 7, it is unclear what the data points represent. The figure legends only indicate that the n = 3/group. The authors should clearly explain in the figure legend. Does each data point represent a region of interest on a lung section? Or perhaps the quantification of an entire lung section?

Reviewer #2

(Remarks to the Author)

The manuscript by Dr. El Agha and colleagues identifies a novel ADAMTS4-positive cells derived from alveolar fibroblasts in pulmonary fibrosis (PF). Authors report enrichment in this cell subpopulation as a persistent cellular sub-cluster during a resolution of PF in aged mouse models and human diseases highlighting a clinical relevance of the study. Multiple state-of-the-art approaches, including scRNA-seq-based lineage tracing, omics, animal models and human diseased data, are used in the study. Authors are also exploring potential therapeutic approaches to mitigate pathophysiological impact of ADAMTS4 upregulation by performing pilot preclinical experiments using rTIMP3, broad MMP inhibitor, in 3D organoid and PCLS models. The study is built on novel Fgf10-Cre-ERT2-tdTomato mice generated in Dr. El Agha's lab. Manuscript reports previously published data from Dr. El Agha team and well integrates novel findings including published studies by other investigative teams. Overall, this study is novel, well-designed, and well-performed with robust statistical analyses with a potential to substantially contribute to pathobiology of pulmonary fibrosis in aging lung.

Major comments:

1. Experimental evidence leading to importance of ADAMS4 in sustained fibroblasts activation induced by bleo in aged mice are presented in Figures 1-3. ADAMTS4 data only appears in Figure 4. It would be easier for a reader to appreciate an importance of the ADAMS4 if data from Figures 1-2 are made more concise and combined in one figure with some data included in Supplemental Data.
2. Figure 5: in organoids experiments, authors should demonstrate expression levels of ADAMS4 and TIMP3 in mesenchymal cells (vehicle-treated) and their reduction in rTIMP3 by western blot or qPCR.
3. Figure 5: in PCLS experiments, authors should demonstrate which cells are expressing ADAMS4 and TIMP3 in mesenchymal cells (vehicle-treated) and their reduction in rTIMP3 by immunohistochemistry.
4. If possible, it would be important to demonstrate in in vitro experiments that rTIMS3 inhibits ADAMS4 metallopeptidase activity.

Reviewer #3

(Remarks to the Author)

Reviewer #4

(Remarks to the Author)

Summary and major comments

This study lineage-traced FGF10+ fibroblasts in aged mice after bleomycin injury. The authors analyzed the dynamic transcriptomic changes of FGF10+ cells and identified a reversible fate transition of lipofibroblasts-to-myofibroblasts during injury and resolution. They found Adamts4 remains highly expressed in the traced fibroblasts during injury and resolution. Multiple ex vivo culture models were employed to investigate the roles of Adamts4 during fibrosis using MMP inhibitors. Moreover, high expression of Adamts4 is identified in human fibrotic lungs and functional assays were performed using human IPF lung explants.

Although the authors successfully modeled irreversible fibrosis using aged mice, the discovery of MyoFB-to-LIF reversing a previous differentiation event is not novel. Moreover, the analysis provided does not fully support this interpretation as the labeled FGF10+ fibroblasts are highly heterogenous and only contain a small fraction of cells with Lipofibroblast features. The ex vivo experiments showed promising results in attenuating fibrosis. However, the lack of specificity of the inhibitor (broad metalloprotease inhibition, rather than specific for Adamts4) diminished the interest of the essays. It would be helpful to use RNAi and/or genetically modified mice to specifically inhibit Adamts4.

Overall, the discovery made by computational analysis is interesting yet with limited novelty. And the ex vivo experiments could not fully support the author's statement that Adamts4 inhibition mitigates fibrosis.

Minor comments:

1. Why are only female mice used in this study? Explanations should be provided.
2. The mouse scRNA-seq analysis in Figure 2 is confusing. More detailed characterization of the fibroblast subtypes would be necessary to help readers understand the subtype differences. For example, what are the key markers of AF1 and AF2? What are the differences between the two LIF-high populations identified in UMAP of AF1? Where are they located in the tissue before and after injury? How do they compare to the AF1/AF2 identified in other studies (eg. [10.1126/science.ado5561](https://doi.org/10.1126/science.ado5561))?
3. More explanation should be provided on using VCAN (versican) in some of the ex vivo assays. Is it used as an additional substrate for Adamts4? Versican was shown to promote fibroblast regeneration (e.g. [10.1161/CIRCULATIONAHA.123.066298](https://doi.org/10.1161/CIRCULATIONAHA.123.066298)). How to confirm the phenotype change observed in this study is through Adamst4-versican axis?
4. Histology of mPCLS and hPCLS in culture would make the analysis more complete.
5. In figure 7, does the n indicate the number of biological samples or technical replicates?
6. Figure 7c is not shown in the figure legend.

Reviewer #5

(Remarks to the Author)

Zabihi et al. have utilized lineage tracing, organoids/PCLS and computational methods to investigate fibroblast lineage relationships during fibrosis and resolution. The data largely corroborate previous studies from this group (and others), and as such deliver an incremental advance in our understanding of fibrosis/resolution – namely that LIF, contained within the FGF10 labeled population, give rise to MyoFB and a persistent ADAMTS4+ lineage. The main novelty provided is the potential pro-fibrotic role for ADAMTS4 which has been reported elsewhere in the context of lung remodeling after influenza infection and cardiac fibrosis. A deeper understanding of the cells that express ADAMTS4 and the mechanism by which its expression are induced would strengthen the impact of this work.

The use of old animals to study age related disease is increasing in popularity. While logical, the authors' use of this model is not substantiated by data. Is there a lack of ADAMTS4 expression in younger bleomycin injured lungs that allows for faster resolution?

The fact that only ~50% of FGF10 lineage traced cells are LIF in saline lungs complicates downstream analysis. Population level dynamics can be assessed, but the authors are overly confident in claims about single cell lineage trajectories. Of specific importance, the authors may conclude that ADAMTS4+ cells derive from an FGF10+ cell, but it is not clear to me whether this is through a MyoFB intermediate, or whether they are only derived from FGF10 + cells. The expression of ADAMTS4 looks nearly ubiquitous in panel 4E. What proportion of ADAMTS4+ cells is negative for tdTomato? The authors should include ADAMTS4 expression data in the larger integrative UMAP.

There is a claim that tdTom+ cells are "massively recruited" (line 250) to areas of fibrosis, but recruitment is not substantiated by data. At best this is an accumulation of tdTom+ cells by unknown mechanism (most likely local proliferation).

The computational data implicate ADAMTS4 in persistent lung remodeling. While supported by data from other studies, the modulators used in the current study for functional studies (e.g., TIMP3) are non-specific. The authors should consider generating complementary genetic data.

Reviewer #6

(Remarks to the Author)

Version 1:

Reviewer comments:

Reviewer #1

(Remarks to the Author)

In the revision of their manuscript, "Persistence of alveolar fibroblast-derived ADAMTS4+ cells during delayed resolution of pulmonary fibrosis," Zabihi et al. have performed extensive additional experimentation and have strengthened support for their conclusions. The authors have clearly addressed each of my previous comments and concerns, and I have no further comments. This study represents an important advance in our understanding of the molecular pathways by which specific fibroblast populations contribute to pulmonary fibrosis.

Reviewer #2

(Remarks to the Author)

Authors addressed all my comments. I have no further questions or concerns.

Reviewer #3

(Remarks to the Author)

Reviewer #4

(Remarks to the Author)

The revised version is much clearer and the new experiments support the authors' hypothesis that ADAMTS4 plays a role in persistent fibrosis.

Minor point:

What are the dotted lines on the UMAPs in Figure S4b,c,h,i supposed to represent? This should be indicated in the figure legend. In both cases, it seems to be a subset of the sub-set but not clear why these cell states have been highlighted.

Reviewer #5

(Remarks to the Author)

The authors have adequately addressed my concerns.

Reviewer #6

(Remarks to the Author)

Point-by-point response letter

Reviewer #1 (Remarks to the Author):

In their manuscript, “Persistence of alveolar fibroblast-derived ADAMTS4+ cells during delayed resolution of pulmonary fibrosis,” Zabihi et al. present an in-depth transcriptomic and tissue imaging analysis of lung mesenchymal states associated with delayed resolution of pulmonary fibrosis in mouse models and with idiopathic pulmonary fibrosis in humans. The authors use an aged mouse model to investigate how Fgf10-lineage traced lung mesenchymal cells determine the kinetics of resolution of pulmonary fibrosis following bleomycin-induced injury. Building on their previous work, the authors found that some of the Fgf10+ mesenchymal cells have a lipofibroblast/AT1 signature and these cells acquire markers associated with myofibroblasts.. During resolution of fibrosis, these lineage-traced cells can interconvert back to a state with a lipofibroblast/AT1 signature. Analysis of the dynamics of Fgf10-lineage traced cells during development and resolution of fibrosis in mouse models found that expression of the gene *Adamts4* was enriched in areas of non-resolving fibrosis. The authors then performed in vitro experiments in lung organoids and ex vivo experiments in precision cut lung slices to test the therapeutic effect of targeting ADAMTS4 activity and one of its known extracellular matrix substrates, versican. To determine the relevance of these findings to human lung health, the authors analyzed lung tissue from IPF patients and precision cut lung slices from humans. They found that lung cells expressing ADAMTS4 and another marker of fibroblasts associated with pulmonary fibrosis, *CTHCR1*, were enriched in tissue from IPF patients compared to controls.

The study provides novel insight into mesenchymal cell states and fibroblast-produced molecules that may prevent resolution of fibrosis following lung injury. The experiments are well designed and the methodology is sound. However, additional analyses are required to determine the fibroblast lineages that give rise to the ADAMTS4+ fibroblasts associated with delayed resolution. Additional experiments are also needed to mechanistically link ADAMTS4 activity to the fibrogenesis observed in in vitro and ex vivo models. Addressing the following major and minor comments would provide further support for the authors’ conclusions and improve the manuscript.

We thank the reviewer for the overall positive evaluation of our work.

Major

Comments:

1. In Figure 1, the authors provide convincing evidence that Fgf10 lineage-traced cells increase in frequency following bleomycin-induced lung injury and that expression of markers associated with a lipofibroblast phenotype (*Tcf21*, *Inmt*, and *Robo2*) decreases among these cells before increasing during injury resolution. In line 250, the authors state that the Fgf10-lineage traced “tdTom+ cells were massively recruited to the fibrotic regions...” Do the authors have any additional evidence that the tdTom+ cells are recruited to the fibrotic regions from other sites as opposed to expanding as a result of local proliferation of these cells? In order to address this, the authors could assess proliferation in the tdTom+ cells or clarify what they mean by “recruitment.”

We thank the reviewer for this important remark. We have mistakenly used the term “recruited” in this instance and have now replaced it with “accumulated”. We also removed the term “massively” since exaggerated language describing the data is generally discouraged according to *Nature Communications* guidelines. To further address this issue, we carried out, as suggested by the reviewer, immunofluorescence for the proliferation marker Ki-67, but we could not detect significant proliferation in tdTom+ cells at day 14 (**Figure 1 for Reviewers; shown below**).

Figure 1 for Reviewers

This is in line with the notion that myofibroblasts do not significantly proliferate at this stage. However, it is clear that there is a significant increase in the abundance of tdTom+ cells as shown in the flow cytometry data (**revised Fig. 2c**), indicating that the cells indeed proliferate early on during the inflammatory phase/early fibrotic phase.

As mentioned above and in order to avoid any misinterpretation of the data, we have corrected this mistake in the relevant section: “RNAscope and immunofluorescence were carried out and the data revealed that tdTom+ cells were **accumulated** in fibrotic regions of Bleo d14 lungs (35.1% ± 2.7%) compared with normal regions of Sal lungs (19.77% ± 1.02%) (Fig. 3a, b, i)”.

2. Related to the comment above, the authors show in Figure 2 that Fgf10 lineage-traced cells, as identified by tdTomato expression, represent a heterogeneous population of mesenchymal cells, including AF1, AF2, and peribronchial fibroblasts. These results appear to be in contrast to the authors statement in the introduction (lines 104 - 107) that lipofibroblasts are characterized by expression of Fgf10. Is the heterogeneity observed in the scRNAseq data a result of an impure population of tdTomato+ due to sorting? Or does the scRNAseq data likely represent the true heterogeneity of Fgf10 lineage-traced cells? In order to address this question, the authors could perform an analysis of their imaging data from the saline condition to determine if the tdTomato+ cells are located in regions of the lung where AF2 and peribronchial fibroblasts would be expected to be observed.

This is another important remark. We would like to clarify that we did not mean to imply by that statement that *Fgf10* is only expressed in lipofibroblasts. Although we and others have shown that lipofibroblasts are marked by *Fgf10* expression, *Fgf10* is also detectable in other cell types in the lung^{1,2}. Unlike epithelial markers that show more stringent expression in defined epithelial cell types along the proximal-distal axis, mesenchymal markers (including *Fgf10*) are not restricted to specific cell types but are rather expressed in multiple mesenchymal subsets. For example, we have previously demonstrated that α SMA (or *Acta2*) expression is not only limited to smooth muscle cells but also marks other adventitial and alveolar populations^{3,4}. In the current scRNA-seq data, we found that the *Fgf10*^{Cre-ERT2}; *tdTomato*^{fllox} approach indeed enriches for lipofibroblasts (AF1) but also labels AF2 and to a lesser extent peribronchial fibroblasts. To remove this ambiguity, we have added the following statement to the relevant section in the Introduction: “It is important to mention that there is no single marker or transgenic line that can currently be used to exclusively label or target LIFs.”

As the reviewer suggested, we carried out RNAscope for top markers of AF2 (*Pi16*) and peribronchial fibroblasts (*Hhip*) in the saline sections and confirmed that a proportion of tdTom+ cells indeed expresses these markers and are located in the corresponding anatomical regions (**New Supplementary Fig. 2a-c**). AF2s, also known as adventitial fibroblasts, are located in the alveolar regions and can also be found in the adventitial cuffs surrounding large airways and arteries (**New Supplementary Fig. 2a**; please note the blood vessel (BV) marked by ACTA2) and as previously described in the atlas of collagen-producing cells⁵. Therefore, we strongly believe that the scRNA-seq data represent real heterogeneity in *Fgf10* lineage-traced cells.

3. Figure 3b includes a heat map representing the differentially expressed genes between AF1 and MyoFB cells across different time points after bleomycin treatment. More details of this analysis are required to properly interpret the data. What does the ‘Expression level’ represent in this analysis? Is it a pairwise comparison between the AF1 cluster and the MyoFB cluster for each of the genes at the indicated time point? If so, I would have expected that markers gene of AF1 cells (and lipofibroblasts), including *Scube2*, *Wnt2*, and *Npnt*, to be enriched in the AF1 cells in the saline condition and not following bleomycin injury at day 14.

We apologize for the lack of clarity in describing these results. The AF1-MyoFB comparison was only done in old Fig. 3a (**new Supplementary Fig. 2f**), thus top AF1 markers such as *Inmt* are shown to be among the top upregulated DEGs in AF1 while ECM genes such as *Sparc* and *Postn* are top markers in MyoFB. The heatmap shown in old Fig. 3b (**new Supplementary Fig. 2g**) displays a pseudo-bulk analysis of the data, thus showing the top DEGs per sample regardless of the constituent cellular clusters. What the data show is that the AF1 signature is highest at d60, which is in line with strong lipofibroblast differentiation and reinforcement during resolution (even compared to saline). Day 14 is dominated by myofibroblasts and day 30 shows gradual downregulation of MyoFB markers and induction of periFB and AF2 markers. “Expression level” was calculated based on Log₂FC of the top 10 genes in each sample. We have added this definition to the corresponding figure legend.

The authors could take advantage of module scoring as they did in Figure 2 h, i, o, and p to assess the dynamics of these different cell states across the time points.

We thank the reviewer for this suggestion. We have now applied the mentioned scoring module to the different timepoints and added the data to **New Supplementary Fig. 3**. The data clearly show that the lipofibroblast (LIF) signature is induced at day 60 compared to the other timepoints (**Supplementary Fig. 3a**) while the Myofibroblast (MYO) signature is induced at day 14 and regresses through d60 (**Supplementary Fig. 3b**).

4. In Figure 5, the authors perform elegant studies modeling aspects of fibrogenesis in mouse lung organoids and precision cut lung slices. They induce gene expression of markers of fibrosis using TGFB and are able to reverse this gene expression by treating the organoids with recombinant TIMP3 or the precision cut lung slices with versican. While both TIMP3 and versican interact with ADAMTS4, there is no direct evidence presented indicating that ADAMTS4 is present and active in these models. Do the authors have any evidence that *Adamts4* is expressed following TGFB treatment or that ADAMTS4 protein is produced and inhibited by TIMP3? Does knocking out or knocking down *Adamts4* gene expression have the same effect as TIMP3 treatment?

We thank the reviewer for this important comment. We have now split old Fig 5 into two (**new Fig. 4** for alveolar organoids and **new Fig. 5** for mouse PCLS) and carried out more detailed analyses. In the organoid setup, we now show that TGFI31 treatment induces *Adamts4* expression, which is attenuated by rTIMP-3 treatment (**new Fig. 4k**). For the PCLS set up, we carried out both qPCR (**new Fig. 5e**) and immunofluorescence for ADAMTS4 (**new Fig. 5h-q**) and we now show that it is indeed upregulated in response to TGFI31 treatment and that this is inhibited by TIMP3. We also purchased an ADAMTS4 activity assay and subjected protein lysates to this assay. As shown in **New Fig. 5g**, TGFI31 significantly enhances ADAMTS4 activity in these tissues, and this is significantly inhibited by rTIMP3.

Regarding gene knockdown of ADAMTS4, we treated human donor PCLS with vehicle or TGFI31, followed by TGFI31 + si-Scrambled or TGFI31 + si-*ADAMTS4*, and analyzed the samples by immunofluorescence, in situ hybridization, qPCR, western blotting, and ADAMTS4 activity assay (**new Fig. 9g-l** and **new Supplementary Fig. 6**). The data show that knocking down *ADAMTS4* inhibited TGFI31-mediated fibrogenesis in human PCLS similarly to rTIMP3 or versican treatment.

Relatedly, in Figures 5 n - o, how do the authors propose that versican is having its effect in reducing expression of these markers of fibrosis? Is the versican directly affecting mesenchymal cell activation or does it interact with ADAMTS4 in a way that it saturates its protease activity?

This is another excellent question. Similarly to the rTIMP-3 experiments, we carried out qPCR on VCAN-treated PCLS, and the results showed that VCAN strongly downregulates *Adamts4* (**new Supplementary Fig. 5b – far right panel**). Whether this effect is direct or indirect remains to be explored. We also carried out immunofluorescence for ADAMTS4 and in situ hybridization for top DEGs of mesenchymal clusters and the results confirmed downregulation of ADAMTS4 at the

protein level by rVCAN treatment (**new Supplementary Fig. 5c-g**) while upregulating *Fgf10* and *Pi16* and downregulating *Cthrc1* (**new Supplementary Fig. 5h-k**). It is also possible that excess VCAN saturates the protease activity of ADAMTS4 or directly affects mesenchymal cell activation (for example in 10.1161/CIRCULATIONAHA.123.066298 as pointed out by Reviewer 4). This exciting debate was added to the revised Discussion as follows:

“Our ex vivo investigations revealed that rTIMP-3 or rVCAN treatment attenuates fibrogenesis at least in part by downregulating *Adamts4* at the RNA and protein levels. To our knowledge, this phenomenon has not been described in the literature so far. *Adamts4* downregulation might be due to a feedback mechanism involving cues originating from the ECM being modulated by rTIMP3 or rVCAN. As expected, activity assays showed reduced ADAMTS4 activity upon rTIMP-3 or siRNA-mediated *ADAMTS4* knockdown. It is also possible that excess rVCAN saturates the protease activity of ADAMTS4, thus attenuating its detrimental effect. Another possibility is that VCAN acts directly on mesenchymal cells similarly to what has been shown in the context of cardiac repair⁶. It might also be that applying excess amounts of intact VCAN dilutes the profibrotic effect of VCAN fragments generated by ADAMTS4-mediated degradation. Thorough analysis of ECM modulation by ADAMTS4, its regulators, and its substrates as well as the associated contribution to sustained lung damage is an important aspect for further research.”

Minor comments:

5. In Figure 2, there is an overwhelming number of analyses presented. Some of these panels could be moved to a supplemental figure. In addition, it would be useful to present the entire gating strategy used to sort tdTomato+ including the forward and side scatter gating along with any gating on a live/dead dye.

This is a well-taken point. We have now revised this figure by including the most important data and introduced ADAMTS4 in **new Fig. 2h-k**. The remaining data were moved to **new Supplementary Fig. 4**. Also, the full gating strategy including live/dead staining and forward and side scatters are now included in **new Fig. 2b**.

6. In Figure 3o, the authors report that ~25% of total cells (among DAPI+) are *Cthrc1*+ in the saline controls as measured by RNA in-situ hybridization. Can the authors comment on whether such a high frequency of cells expressing *Cthrc1* was expected prior to bleomycin injury?

This is another excellent observation. We agree that *Cthrc1* is often assumed to be exclusively expressed in the fibrotic setting; however, we and others have shown that it is readily expressed in mesenchymal cells during homeostasis, but is significantly upregulated upon fibrotic injury^{4,5}.

7. The authors use an analysis of ‘inferred latent time’ in several figures in the manuscript (4a, S2a). For these analyses, what does the color-coding indicate?

The color-coding indicates latent time. We have now corrected this in **new Fig. 2h**.

8. There is a minor typo in line 268 - change “ADATMS4” to “ADAMTS4”

We have corrected this typo.

9. For the quantification of the immunofluorescence imaging in Figures 1, 3, 4, and 7, it is unclear what the data points represent. The figure legends only indicate that the $n = 3/\text{group}$. The authors should clearly explain in the figure legend. Does each data point represent a region of interest on a lung section? Or perhaps the quantification of an entire lung section?

We apologize for the lack of clarity. Each data point represents the average of several sections from one biological replicate (one lung). Therefore, each data point represents one independent lung. We have added this information to the corresponding figure legends.

Reviewer #2 (Remarks to the Author):

The manuscript by Dr. El Agha and colleagues identifies a novel ADAMTS4-positive cells derived from alveolar fibroblasts in pulmonary fibrosis (PF). Authors report enrichment in this cell subpopulation as a persistent cellular sub-cluster during a resolution of PF in aged mouse models and human diseases highlighting a clinical relevance of the study. Multiple state-of-the-art approaches, including scRNA-seq-based lineage tracing, omics, animal models and human diseased data, are used in the study. Authors are also exploring potential therapeutic approaches to mitigate pathophysiological impact of ADAMTS4 upregulation by performing pilot preclinical experiments using rTIMP3, broad MMP inhibitor, in 3D organoid and PCLS models. The study is built on novel Fgf10-Cre-ERT2-tdTomato mice generated in Dr. El Agha's lab. Manuscript reports previously published data from Dr. El Agha team and well integrates novel findings including published studies by other investigative teams. Overall, this study is novel, well-designed, and well-performed with robust statistical analyses with a potential to substantially contribute to pathobiology of pulmonary fibrosis in aging lung.

We thank the reviewer for the overall positive feedback.

Major comments:

1. Experimental evidence leading to importance of ADAMTS4 in sustained fibroblasts activation induced by bleo in aged mice are presented in Figures 1-3. ADAMTS4 data only appears in Figure 4. It would be easier for a reader to appreciate an importance of the ADAMTS4 if data from Figures 1-2 are made more concise and combined in one figure with some data included in Supplemental Data.

We thank the reviewer for this insightful remark. We have now rearranged the figures so that ADAMTS4 appears directly in the first scRNA-seq figure (**new Fig. 2h-k**) instead of old Fig. 4 and we moved the "less relevant" data to **new Supplementary Figures 2-4**. The in situ hybridization for ADAMTS4 that was previously in Fig. 4 is now in **new Fig. 3**. We hope that these changes address the reviewer's comment and enhance the focus, read quality, and logical flow of the paper.

2. Figure 5: in organoids experiments, authors should demonstrate expression levels of ADAMTS4 and TIMP3 in mesenchymal cells (vehicle-treated) and their reduction in rTIMP3 by western blot or qPCR.

For the revision of the paper, we carried out qPCR for *Adamts4* and *Timp3* on the alveolar organoids as suggested by the reviewer, and we now show that *Adamts4* is upregulated by rTGFβ1 and that this is largely attenuated by rTIMP-3 treatment (**new Fig. 4k**). On the other hand, *Timp3* did not show significant changes (**new Fig. 4l**). Since the organoids were treated with an excess of exogenous recombinant TIMP3, we do not believe that the endogenous levels of *Timp3* expression have a significant impact on ADAMTS4 activity.

3. Figure 5: in PCLS experiments, authors should demonstrate which cells are expressing ADAMTS4 and TIMP3 in mesenchymal cells (vehicle-treated) and their reduction in rTIMP3 by immunohistochemistry.

As requested by the reviewer, we carried out immunofluorescence for ADAMTS4 combined with in situ hybridization for selected DEGs defining the different mesenchymal clusters on treated PCLS (*Tcf21* for AF1, *Pi16* for AF2, *Hhip* for PeriFB, and *Cthrc1* for MyoFB), and the results showed that ADAMTS4 is indeed upregulated at the protein level by TGF β 1 treatment and that this is inhibited by rTIMP-3 treatment (**new Fig. 5h-u**). The data also show that rTGFJ31 downregulates *Tcf21* (AF1 marker) (**Fig. 5r**) while inducing *Pi16* (AF2) (**Fig. 5s**), *Cthrc1* (MyoFB) (**Fig. 5t**), and to some extent *Hhip* (PeriFB) (**Fig. 5u**), which was largely reversed by rTIMP-3 treatment. Recombinant TGFJ31 induced the expression of ADAMTS4 mostly in AF1 (**Fig. 5r**) and AF2 (**Fig. 5s**), which was reversed by rTIMP-3, especially in AF1.

Similarly to the organoid data, we also carried out qPCR for *Adamts4* and *Timp3* and the results were similar. *Adamts4* is upregulated by rTGF β 1, and this is largely attenuated by rTIMP-3 treatment (**new Fig. 5e**). *Timp3* did not show significant changes across groups (**new Fig. 5f**). We also investigated the mouse scRNA-seq dataset and found that *Timp3* is ubiquitously expressed across clusters in contrast to *Adamts4* that is significantly more restricted (**Figure 2 for Reviewers; shown below**). Since *Timp3* is ubiquitously expressed across clusters (**Figure 2 for Reviewers; shown below**) without significant regulation across groups (**new Fig. 5f**), and since the endogenous expression of *Timp3* is unlikely to have a significant impact on ADAMTS4 activity in the presence of exogenously applied recombinant TIMP-3, we feel that carrying out the immunofluorescence in combination with the in situ hybridization would not provide additional insight in this context. However, we sincerely appreciate the reviewer's suggestion and kindly ask for their understanding regarding this limitation.

Figure 2 for Reviewers

4. If possible, it would be important to demonstrate in in vitro experiments that rTIMP3 inhibits ADAMTS4 metalloproteinase activity.

This is another excellent remark. For the revision of the paper, we obtained a commercially available ADAMTS4 activity assay and carried out the requested measurements. As shown in **new Fig. 5g**, treatment with rTIMP-3 indeed inhibits ADAMTS4 activity that is induced by TGF131 treatment. We also carried out this activity assay on human PCLS treated with TGF131 followed by siRNA against *ADAMTS4* and the data show significant increase in response to TGF131 treatment and subsequent decrease upon gene knockdown (**new Fig. 9I**).

Reviewer #3 (Remarks to the Author):

We thank the reviewer for co-reviewing the paper.

Reviewer #4 (Remarks to the Author):

Summary and major comments

This study lineage-traced FGF10+ fibroblasts in aged mice after bleomycin injury. The authors analyzed the dynamic transcriptomic changes of FGF10+ cells and identified a reversible fate transition of lipofibroblasts-to-myofibroblasts during injury and resolution. They found *Adamts4* remains highly expressed in the traced fibroblasts during injury and resolution. Multiple ex vivo culture models were employed to investigate the roles of *Adamts4* during fibrosis using MMP inhibitors. Moreover, high expression of *Adamts4* is identified in human fibrotic lungs and functional assays were performed using human IPF lung explants.

1. Although the authors successfully modeled irreversible fibrosis using aged mice, the discovery of MyoFB-to-LIF reversing a previous differentiation event is not novel. Moreover, the analysis provided does not fully support this interpretation as the labeled FGF10+ fibroblasts are highly heterogeneous and only contain a small fraction of cells with Lipofibroblast features.

We thank the reviewer for the feedback. We appreciate that certain aspects of MyoFB-to-LIF differentiation during fibrosis resolution have been reported by us and others; however, our current study is the first to provide proof and detailed analysis of such event using a driver line that enriches for LIF (around 55% of labeled cells are LIFs at steady state as determined by histology and single-cell transcriptomics) and in the context of delayed fibrosis resolution in aged mice. This approach allowed the identification of *ADAMTS4*+ cells as a novel progeny of FGF10+ cells during delayed fibrosis resolution. For the revision of the paper, we carried out additional bioinformatic analysis and the data show that *ADAMTS4*+ cells are mainly contained in the AF1 (lipofibroblast) cluster (**new Fig. 2i-k**) and RNA velocity analysis showed that they derive from AF1 cells (**new Fig. 2g**). Therefore, despite the expected heterogeneity of FGF10+ cells, the findings are strongly linked to the lipofibroblast cluster. To further address this concern, we also added the following statement to the Introduction: "It is important to mention that there is no single marker or transgenic line that can currently be used to exclusively label or target LIFs."

2. The ex vivo experiments showed promising results in attenuating fibrosis. However, the lack of specificity of the inhibitor (broad metalloprotease inhibition, rather than specific for *Adamts4*) diminished the interest of the essays. It would be helpful to use RNAi and/or genetically modified mice to specifically inhibit *Adamts4*.

We thank the reviewer for the important comment. For the revision of this paper, we treated human donor PCLS with vehicle or TGF131, followed by TGF131 + si-Scrambled or TGF131 + si-*ADAMTS4*, and analyzed the samples by immunofluorescence, in situ hybridization, qPCR, western blotting, and *ADAMTS4* activity assay (**new Fig. 9g-l and new Supplementary Fig. 6**). The data show that knocking down *ADAMTS4* inhibited TGF131-mediated fibrogenesis in human PCLS similarly to rTIMP3 or versican treatment.

We would like to highlight that the computational analysis using the mouse model allowed the identification of persistent ADAMTS4+ cells in delayed fibrosis resolution. In addition to the single-cell RNA-seq data from the human lung cell atlas showing that *ADAMTS4* is among the top upregulated genes in pulmonary fibrosis, we have now added spatial transcriptomic data (**new Fig. 8**) to consolidate this finding. Further validation using ex vivo approaches, especially the new data utilizing siRNA against *ADAMTS4* on human PCLS (**new Fig. 9g-l**) indicate strong clinical relevance. We hope that the revisions that we made alleviate the reviewer's concern.

Minor comments:

1. Why are only female mice used in this study? Explanations should be provided.

Aged female mice were used because they show better survival and less severe lung fibrosis in response to bleomycin instillation compared with aged male mice⁷. Since the aim of the study was to study fibrosis resolution, female mice were chosen for the experiments. We have added this explanation to the relevant Methods section of the revised manuscript as follows: "Since aged female mice show better survival and less severe lung fibrosis in response to bleomycin instillation compared with aged male mice⁷, they were chosen for the experiments."

2. The mouse scRNA-seq analysis in Figure 2 is confusing. More detailed characterization of the fibroblast subtypes would be necessary to help readers understand the subtype differences. For example, what are the key markers of AF1 and AF2? What are the differences between the two LIF-high populations identified in UMAP of AF1? Where are they located in the tissue before and after injury? How do they compare to the AF1/AF2 identified in other studies (eg. [10.1126/science.ado5561](https://doi.org/10.1126/science.ado5561))?

We thank the reviewer for the important feedback. We have reworked this part of the manuscript and rearranged the data and figures. We now provide the top 10 markers of AF1 and AF2 in **new Fig. 2f** and show in situ analysis of AF2 in **new Supplementary Fig. 2a**. In situ analysis of AF1 is shown in **Fig. 1**.

Regarding the two LIF-high populations (clusters 3 and 5), the cluster frequency plot shows that cluster 5 (red) is the abundant one during homeostasis and cluster 3 (green) becomes abundant during injury and resolution (**new Supplementary Fig. 4h, i, l**). To address the comment raised by the reviewer, we selected the top 500 DEGs for each of the two clusters and subjected them to GO analysis (**Figure 3 for Reviewers; shown below**). The data show comparison between cluster 3 vs. 5 at each timepoint/condition. The take-home message is that cluster 3 is enriched for regulation of apoptosis in response to DNA damage in the saline condition as well as programs such as ECM organization and remodeling and mesenchyme development during active fibrosis

and resolution. These data agree with the notion that cluster 3 (enriched for *Scube2*, *Ces1d*, *Tcf21*, *Wnt2*; please also see Supplementary Data file) is the injury-induced LIF cluster.

Figure 3 for Reviewers

Regarding the spatial distribution of these two clusters, we datamined a published spatial transcriptomics dataset. We first performed a module score based on the top 100 genes for each of the two clusters and then plotted them as spatial plots on control and day 21 bleo (datamined from ⁸ (Figure 4 for Reviewers; shown below). In agreement with our scRNA-seq data, Cluster 5 (red; steady state LIF-high) is downregulated after bleo while cluster 3 (green; injury-induced LIF-high) is enriched in remodeled regions (see Factor 10 – ECM remodeling). To address the

difference between these two clusters in the manuscript, we added the following statement to the relevant passage in the Results section: “The two *LIF^{high}* subclusters correspond to steady-state LIFs (red) and injury-induced LIFs (green). Injury-induced LIFs are enriched for regulation of apoptosis in response to DNA damage in the saline condition and programs such as ECM organization and remodeling and mesenchyme development during active fibrosis and resolution.”

Figure 4 for Reviewers

As to how these two populations compare to AF1 and AF2 described in 10.1126/science.ado5561 ⁹, we integrated our scRNA-seq dataset with the *Pdgfra*+ dataset from that study, and we found that our AF1s (AF1_Fgf10_Bleo) positively correlate with their AF1s (AF1_Pdgfra_Bleo) as do our AF2s (AF2_Fgf10_Bleo) with their AF2 (AF2_Pdgfra_Bleo) (**Purple boxes in Figure 5 for Reviewers;**

shown below). Therefore, our two LIF-high clusters are contained in AF1 but not AF2 reported by Jones et al.

Figure 5 for Reviewers

3. More explanation should be provided on using VCAN (versican) in some of the ex vivo assays. Is it used as an additional substrate for Adamts4? Versican was shown to promote fibroblast regeneration (e.g. 10.1161/CIRCULATIONAHA.123.066298). How to confirm the phenotype change observed in this study is through Adamts4-versican axis?

We thank the reviewer for the important remark. For the revision of the paper, we carried out qPCR and immunofluorescence for ADAMTS4 in VCAN-treated PCLS, and the results showed that VCAN strongly downregulates ADAMTS4 in mouse and human PCLS (new Supplementary Fig. 5b, g, new Fig. 9a-b, and new Supplementary Fig. 9a-f). To our knowledge, this phenomenon has not been described in the literature so far. ADAMTS4 downregulation might be due to a feedback mechanism involving cues originating from the ECM that is being modulated by excess VCAN. It

is also possible that excess VCAN saturates the protease activity of ADAMTS4 or directly affects mesenchymal cell activation (for example in 10.1161/CIRCULATIONAHA.123.066298 as pointed out by the reviewer). Since versican is a major substrate for ADAMTS4 and the siRNA-mediated knockdown of *ADAMTS4* (**new Fig. 9g-l**) reproduced the effect seen with rTIMP-3 or rVCAN treatment, we think that the phenotype change is through the ADAMTS4-VCAN axis. This exciting debate was added to the revised Discussion as follows:

“Our ex vivo investigations revealed that rTIMP-3 or rVCAN treatment attenuates fibrogenesis at least in part by downregulating *Adamts4* at the RNA and protein levels. To our knowledge, this phenomenon has not been described in the literature so far. *Adamts4* downregulation might be due to a feedback mechanism involving cues originating from the ECM being modulated by rTIMP3 or rVCAN. As expected, activity assays showed reduced ADAMTS4 activity upon rTIMP-3 or siRNA-mediated *ADAMTS4* knockdown. It is also possible that excess rVCAN saturates the protease activity of ADAMTS4, thus attenuating its detrimental effect. Another possibility is that VCAN acts directly on mesenchymal cells similarly to what has been shown in the context of cardiac repair⁶. It might also be that applying excess amounts of intact VCAN dilutes the profibrotic effect of VCAN fragments generated by ADAMTS4-mediated degradation. Thorough analysis of ECM modulation by ADAMTS4, its regulators, and its substrates as well as the associated contribution to sustained lung damage is an important aspect for further research.”

4. Histology of mPCLS and hPCLS in culture would make the analysis more complete.

We now provide histology data showing in situ hybridization for markers of mesenchymal subsets and ADAMTS4 expression in **new Fig. 5** (mPCLS ± TGFI31 ± rTIMP-3), **new Supplementary Fig. 5** (Bleo mPCLS ± rVCAN), **new Fig. 9c-e** (human IPF PCLS ± rTIMP-3 or rVCAN), **new Fig. 9h** (human donor PCLS ± TGFI31 ± siRNA) and **new Supplementary Fig. 6** (human donor PCLS ± TGFI31 ± siRNA).

5. In figure 7, does the n indicate the number of biological samples or technical replicates?

In Figure 7 (**new Fig. 6**), the n indicates biological samples (each dot represents data from one patient). This information has been added to the figure legend.

6. Figure 7c is not shown in the figure legend.

We have corrected this mistake in the revised manuscript.

Reviewer #5 (Remarks to the Author):

Zabihi et al. have utilized lineage tracing, organoids/PCLS and computational methods to investigate fibroblast lineage relationships during fibrosis and resolution. The data largely corroborate previous studies from this group (and others), and as such deliver an incremental advance in our understanding of fibrosis/resolution – namely that LIF, contained within the FGF10 labeled population, give rise to MyoFB and a persistent ADAMTS4+ lineage. The main novelty provided is the potential pro-fibrotic role for ADAMTS4 which has been reported elsewhere in the context of lung remodeling after influenza infection and cardiac fibrosis. A deeper understanding of the cells that express ADAMTS4 and the mechanism by which its expression are induced would strengthen the impact of this work.

We thank the reviewer for the important feedback.

1. The use of old animals to study age related disease is increasing in popularity. While logical, the authors' use of this model is not substantiated by data. Is there a lack of ADAMTS4 expression in younger bleomycin injured lungs that allows for faster resolution?

(Redacted)

(Redacted)

2. The fact that only ~50% of FGF10 lineage traced cells are LIF in saline lungs complicates downstream analysis. Population level dynamics can be assessed, but the authors are overly confident in claims about single cell lineage trajectories. Of specific importance, the authors may conclude that ADAMTS4+ cells derive from an FGF10+ cell, but it is not clear to me whether this is through a MyoFB intermediate, or whether they are only derived from FGF10 + cells.

We thank the reviewer for the insightful comment. We agree that we cannot be fully certain whether the MyoFB is an obligatory intermediate for FGF10+ cells before they give rise to ADAMTS4+ cells. However, double in situ hybridization for *Cthrc1* and *Adamts4* does not show significant increase in colocalization of these two markers in tdTom+ cells after bleo injury (**new Fig. 3p**). Moreover, RNA velocity analysis predicts a trajectory toward ADAMTS4+ cells within the AF1 cluster without necessarily passing through the MyoFB cluster (**new Fig. 2g**). This leads us to think that FGF10+ cells do not necessarily pass through a MyoFB intermediate before giving rise to *Adamts4*+ cells.

To further address this aspect, we revisited the RNA velocity and latent time trajectories for the revision of the paper. **Figure 7 for Reviewers (shown below)** reveals all ADAMTS4+ cells that derive from the AF1 cluster and the possible paths for their emergence: Path 1 (red) is through the MyoFB state while Path 2 (blue) does not involve the MyoFB state. The analysis indicates that AF1 cells with *Adamts4*+ as a terminal fate do not universally pass through a MyoFB-like

intermediate (mostly Path 2 rather than Path 1). While some transitional states exist, the majority of AF1 cells reach the *Adamts4*⁺ fate without requiring a MyoFB state. Therefore, we think that AF1 → ADAMTS4⁺ differentiation can occur independently of the MyoFB intermediate state.

Figure 7 for Reviewers

However, and as the reviewer pointed out, we acknowledge that the bioinformatic prediction requires experimental validation. Therefore, we added the following statement to the revised Discussion: “While bioinformatic analysis predicted that *Fgf10*⁺ cells do not necessarily pass through a MyoFB intermediate before giving rise to *Adamts4*⁺ cells, this intriguing aspect remains an open question that requires further experimental testing.”

3. The expression of ADAMTS4 looks nearly ubiquitous in panel 4E. What proportion of ADAMTS4⁺ cells is negative for tdTomato?

We thank the reviewer for this comment, which is well-taken. The data in old Fig. 4E (**new Fig. 3h**) shows in situ hybridization for *Adamts4* overlaid with immunofluorescence for tdTomato. Due to the nature of the in situ signal (dots revealing the presence of RNA transcripts) and the high cellular density in the unresolved damaged regions, the expression pattern naturally looks widespread. *Adamts4* expression has been shown to be restricted to fibroblasts and endothelial cells in the mouse lung¹⁰. To further address this comment, we carried out additional quantification and the data show that tdTom⁺ cells represent up to 33% of total *Adamts4*⁺ cells at day 60. While this indicates that there are other sources of ADAMTS4 in the lung (other fibroblasts that are not derived from the FGF10⁺ lineage in addition to endothelial cells), it also

emphasizes the important role played by ADAMTS4 in delayed fibrosis resolution, thus further reinforcing it as a promising therapeutic target. We have added the quantification mentioned above in **new Fig. 3q**.

4. The authors should include ADAMTS4 expression data in the larger integrative UMAP.

As requested, *Adamts4* UMAPs are now provided in the larger integrative UMAP of the main scRNA-seq figure (**Fig. 2i, j**). The UMAPs also show the expression of *Scube2* (AF1), *Pi16* (AF2), and *Cthrc1* (MyoFB).

5. There is a claim that tdTom+ cells are “massively recruited” (line 250) to areas of fibrosis, but recruitment is not substantiated by data. At best this is an accumulation of tdTom+ cells by unknown mechanism (most likely local proliferation).

We thank the reviewer for spotting this mistake. The term “recruited” was replaced by “accumulated” in the relevant passage (**Please also see response to Comment 1 by Reviewer 1**). We also removed the term “massively” as the reviewer also suggested and since exaggerated language describing the data is generally discouraged according to *Nature Communications* guidelines.

4. The computational data implicate ADAMTS4 in persistent lung remodeling. While supported by data from other studies, the modulators used in the current study for functional studies (e.g., TIMP3) are non-specific. The authors should consider generating complementary genetic data.

We thank the reviewer for the important comment. For the revision of this paper, we treated human donor PCLS with vehicle or TGF β 1, followed by TGF β 1 + si-Scrambled or TGF β 1 + si-*ADAMTS4*, and analyzed the samples by immunofluorescence, in situ hybridization, qPCR, western blotting, and ADAMTS4 activity assay (**new Fig. 9g-l and new Supplementary Fig. 6**). The data show that knocking down *ADAMTS4* inhibits TGF β 1-mediated fibrogenesis in human PCLS similarly to rTIMP3 or versican treatment.

Reviewer #6 (Remarks to the Author):

We thank the reviewer for co-reviewing the paper.

References

1. Al Alam, D. *et al.* Evidence for the involvement of fibroblast growth factor 10 in lipofibroblast formation during embryonic lung development. *Dev. Camb. Engl.* **142**, 4139–50 (2015).
2. El Agha, E. *et al.* Fgf10-positive cells represent a progenitor cell population during lung development and postnatally. *Dev. Camb. Engl.* **141**, 296–306 (2014).
3. Khadim, A. *et al.* Myofibroblasts emerge during alveolar regeneration following influenza-virus-induced lung injury. *Cell Rep.* **44**, 115248 (2025).
4. Lingampally, A. *et al.* Evidence for a lipofibroblast-to-Cthrc1 + myofibroblast reversible switch during the development and resolution of lung fibrosis in young mice. *Eur. Respir. J.* **65**, 2300482 (2025).
5. Tsukui, T. *et al.* Collagen-producing lung cell atlas identifies multiple subsets with distinct localization and relevance to fibrosis. *Nat. Commun.* **11**, 1920 (2020).
6. Feng, J. *et al.* Versican Promotes Cardiomyocyte Proliferation and Cardiac Repair. *Circulation* **149**, 1004–1015 (2024).
7. Redente, E. F. *et al.* Age and sex dimorphisms contribute to the severity of bleomycin-induced lung injury and fibrosis. *Am. J. Physiol. Lung Cell. Mol. Physiol.* **301**, L510-518 (2011).
8. Franzén, L. *et al.* Mapping spatially resolved transcriptomes in human and mouse pulmonary fibrosis. *Nat. Genet.* **56**, 1725–1736 (2024).
9. Jones, D. L. *et al.* An injury-induced mesenchymal-epithelial cell niche coordinates regenerative responses in the lung. *Science* **386**, eado5561 (2024).
0. Boyd, D. F. *et al.* Exuberant fibroblast activity compromises lung function via ADAMTS-4. *Nature* **587**, 466–471 (2020).

Response letter

Reviewer #1 (Remarks to the Author):

In the revision of their manuscript, “Persistence of alveolar fibroblast-derived ADAMTS4+ cells during delayed resolution of pulmonary fibrosis,” Zabihi et al. have performed extensive additional experimentation and have strengthened support for their conclusions. The authors have clearly addressed each of my previous comments and concerns, and I have no further comments. This study represents an important advance in our understanding of the molecular pathways by which specific fibroblast populations contribute to pulmonary fibrosis.

We thank the reviewer for the positive evaluation

Reviewer #2 (Remarks to the Author):

Authors addressed all my comments. I have no further questions or concerns.

We thank the reviewer for the positive evaluation

Reviewer #3 (Remarks to the Author):

We thank the reviewer for the positive evaluation

Reviewer #4 (Remarks to the Author):

The revised version is much clearer and the new experiments support the authors' hypothesis that ADAMTS4 plays a role in persistent fibrosis.

We thank the reviewer for the positive evaluation

Minor point:

What are the dotted lines on the UMAPs in Figure S4b,c,h,i supposed to represent? This should be indicated in the figure legend. In both cases, it seems to be a subset of the sub-set but not clear why these cell states have been highlighted.

We thank the reviewer for this remark. The dotted lines in b), c), and d) mark the subclusters with high MyoFB score and those in e), h), i), and j) mark the subclusters with high LIF score. The dotted line in k) highlights the MyoFB score in the *LIFAdamts4* subcluster. We have added this explanation to the corresponding figure legend.

Reviewer #5 (Remarks to the Author):

The authors have adequately addressed my concerns.

We thank the reviewer for the positive evaluation

Reviewer #6 (Remarks to the Author):

We thank the reviewer for the positive evaluation